# Conformational dynamics of free and membrane-bound human Hsp70 in model cytosolic and endo-lysosomal environments

Valeria Calvaresi[1], Line T. Truelsen[1], Sidsel B. Larsen[1], Nikolaj H. T. Petersen[2], Thomas Kirkegaard [2] & Kasper D. Rand [1✉]

The binding of the major stress-inducible human 70-kDa heat shock protein (Hsp70) to the anionic phospholipid bis-(monoacylglycero)-phosphate (BMP) in the lysosomal membrane is crucial for its impact on cellular pathology in lysosomal storage disorders. However, the conformational features of this protein-lipid complex remain unclear. Here, we apply hydrogen–deuterium exchange mass spectrometry (HDX-MS) to describe the dynamics of the full-length Hsp70 in the cytosol and its conformational changes upon translocation into lysosomes. Using wild-type and W90F mutant proteins, we also map and discriminate the interaction of Hsp70 with BMP and other lipid components of the lysosomal membrane. We identify the N-terminal of the nucleotide binding domain (residues 87–118) as the primary orchestrator of BMP interaction. We show that the conformation of this domain is significantly reorganized in the W90F mutant, explaining its inability to stabilize lysosomal membranes. Overall, our results reveal important new molecular details of the protective effect of Hsp70 in lysosomal storage diseases, which, in turn, could guide future drug development.

[1] Protein Analysis Group, Department of Pharmacy, University of Copenhagen, 2100 Copenhagen O, Denmark. [2] Orphazyme A/S, Copenhagen, Denmark. ✉email: kasper.rand@sund.ku.dk

The 70-kDa heat shock proteins (Hsp70s) are molecular chaperones involved in protein folding and cellular homeostasis[1,2]. The major human stress-inducible Hsp70, encoded by HspA1A gene, is expressed in response to cellular stress[3–5] and is involved in tumorigenesis[3,5–7]. Hsp70s are formed of two domains connected by a linker: the nucleotide-binding domain (NBD) and the substrate-binding domain (SBD). The SBD interacts with short linear motifs of unfolded protein substrates. Through the energy provided by ATP hydrolysis in the NBD, an allosteric mechanism induces conformational changes in the substrate-binding cavity, enabling the protein binding and refolding[1,8–18]. Hsp70 is commonly known to interact with protein unfolded substrates in the cytosol. However, Hsp70 can also associate to lipids[19,20], including the anionic phospholipid bis-(monoacylglycero)-phosphate (BMP)[21–24], particularly enriched in the membrane of the intraluminal vesicles of the endo-lysosomal compartment[25,26], where Hsp70 can localize. In vivo, particularly in response to various stresses, Hsp70 and other heat shock proteins are known to be released in the extracellular space, where they elicit various biological responses facilitated by their binding to a number of receptors from different families, such as the toll-like receptors (TLR), scavenger receptors and c-type lectins[27–29]. A cellular uptake mechanism for extracellular Hsp70, by which it may enter the endo-lysosomal compartment, has been demonstrated for recombinant Hsp70 through its interaction with the scavenger receptor LRP-1/CD91[30,31]. In the lysosomal compartment, by associating to BMP of the intraluminal vesicle membranes, Hsp70 prevents lysosomal membrane permeabilization, inhibiting the uncontrolled release of degradative enzymes and consequent cell death[24,32,33]. Accordingly, Hsp70 has shown to revert the lysosomal dysfunctional phenotype of fibroblasts from patients affected by lysosomal storage disorders (LSDs)[24,30,34]. LSDs are rare orphan diseases caused by mutations in acid hydrolases, such as the acid sphingomyelinase[35], or proteins necessary for lysosomal metabolism, such as NPC1[36,37]. LSDs currently lack effective treatment and Hsp70 has been proposed as a therapeutic option[24,30,34,36,38]. The ability of Hsp70 to bind BMP-containing vesicles has been assessed in the pH range 4.5–7.4[21–24], revealing a pH-dependent association, enhanced under acidic pH conditions[23,24]. The binding of Hsp70 to BMP has been studied with a form of this phospholipid containing monounsaturated fatty acid chains (18:1 Δ9). The chaperone does not integrate into lipid bilayers containing BMP in this form, rather associates peripherally[21,24]. The complex is almost totally disrupted with a high salt concentration, indicating that electrostatic forces are important contributors to the binding, although insufficient to fully explain it[23]. Moreover, the addition of a histidine-tag at the N-terminus of Hsp70 influences its lipid-binding properties[21]. It has been demonstrated that the binding to BMP is mainly mediated by the NBD, albeit also the SBD is involved in the interaction[21,22,24]. In addition, it has been shown that the Hsp70 W90F mutant poorly binds to BMP and is unable to revert the phenotype associated with Niemann-Pick diseases[24]. However, we lack a clear molecular understanding of the interaction of Hsp70 to BMP and other lysosomal lipids, and of the conformational and mechanistic consequences of the binding. Indeed, the association of Hsp70 to lysosomes has only been dissected at the domain level, and insufficient local information on residues/regions structurally and functionally implicated in the formation of this complex is available, limiting drug development strategies.

Hydrogen-deuterium exchange coupled to mass spectrometry (HDX-MS) is a sensitive method to assess the conformational dynamics of proteins in solution and in membrane environments[39–43]. Upon incubation in deuterated buffers, protein backbone amides exchange their hydrogen with deuterium

(HDX) and the rate at which this occurs is a highly sensitive reporter of local structure and dynamics[44,45]. Indeed, high % deuteration after a brief exposure time indicates that a given region has high flexibility in solution. This is generally seen for loops and regions of random coil structure. Conversely, folded regions generally show a significantly low % deuteration due to the presence of stable hydrogen-bonding networks. Furthermore, changes in HDX induced by the presence of a ligand can pinpoint protein regions directly or indirectly involved in the binding at resolution of peptide segments[46].

Here we have applied HDX-MS to elucidate the molecular details of the interaction between Hsp70 and the lysosomal membrane. We first compared the conformational dynamics of Hsp70 in a cytosolic (neutral) and lysosomal (acidic) environment, to detect pH-induced conformational changes and explain its low affinity for BMP at neutral pH. We then assessed the association of Hsp70 to liposomes (large unilamellar vesicles, LUVs) made of the most common lysosomal lipids, in an acidic pH range (4.5-5.3) mimicking that of early to mature lysosomes. We mapped the interaction to liposomes of the Hsp70 wild-type (Hsp70 WT), of the N-terminal hexahistidine-tagged Hsp70 (His6-Hsp70 WT) and of the inactive hexahistidine-tagged mutant W90F (His6-Hsp70 W90F). In order to discriminate BMP-specific binding effects, we measured the HDX of the proteins alone and in complex to liposomes with or without BMP. Our results allow us to determine (a) the regions in Hsp70 involved in binding the anionic BMP or zwitterionic lipids, (b) the conformational differences between the wild-type and the binding-incompetent mutant, and (c) that Hsp70 transitions into a molten globular state at the pH of mature lysosomes. Taken together, our findings provide mechanistic insights into how Hsp70 stabilizes the lysosomes and can open up for new therapeutic strategies for LSD treatment.

## Results
**Hsp70 conformation becomes more flexible and dynamic at endo-lysosomal pH**. To understand the solution-phase conformational properties of full-length Hsp70, we primarily optimized labelling and quench buffer conditions for maximizing sequence coverage, also in the presence of lipids (Supplementary Notes 1–3, Supplementary Data 1–3 and Fig. S1). To assess the conformational changes of Hsp70 upon translocation from the cytosol into the lysosomes, we first measured the HDX of Hsp70 at pH 4.5 and 5. Under these buffer conditions, we observed clear indications that the chaperone undergoes irreversible unfolding, as revealed by the presence of bimodal-shaped isotopic envelopes in the mass spectra of peptides spanning most regions of the protein (Supplementary Note 4 and Figs. S2–S5). To study this further, we performed a series of pulse-labelling HDX experiments, which confirmed that the NBD and most of the SBD undergo unfolding to a molten globule-like state at pH of mature lysosomes, on a timescale of a few minutes (Figs. S6–S9). Conversely, at pH 5.3 and 5.5, the structure of Hsp70 remained folded and stable over time; no evidence of irreversible unfolding could be observed (Supplementary Note 5, Figs. S10 and S11). We thus chose to compare the HDX of Hsp70 at pH 7.4 and 5.3, the latter corresponding to the lumen pH of early lysosomes[47]. We measured its HDX by monitoring the deuterium incorporation of 126 peptides, spanning 96.7% of the protein sequence (Fig. S12). The HDX of Hsp70 WT at pH 7.4 varied considerably across different regions. While no crystal structure exists of full-length human Hsp70, a comparison of our HDX data to the structures of the isolated NBD[8] and SBD[9] revealed a good overall agreement (Fig. S13). Generally, fast HDX (defined here as more than 80% of uptake relative to the maximally labelled state after 1 min of

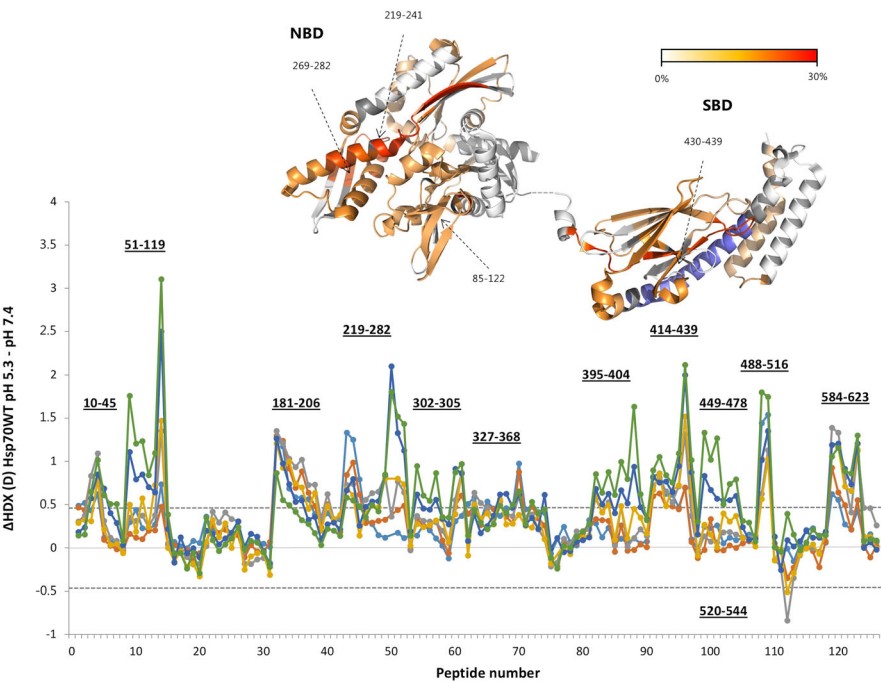

**Fig. 1 Comparison of HDX of His[6]-Hsp70 WT at pH 5.3 and 7.4.** Difference plot illustrating the difference in HDX between pH 5.3 and 7.4 over the measured time points (cyan line: 5 s, orange line: 15 s, grey line: 1 min, yellow line: 2 min, blue line: 5 min, green line: 10 min). Residues comprising a region with significant differences in HDX are indicated. The peptides are arranged according to their position from N- to C-terminus (Supplementary Data 5, Table 2). A dotted grey line indicates the 98% CI as a threshold for significance. Regions showing a significant increased HDX are coloured in a red scale in the crystal structures according to their difference in relative fractional uptake normalized to the uptake of the maximally labelled state (the time point showing higher difference is considered for colouring). The region showing protection toward HDX is coloured in blue. In grey, regions for which no HDX information was retrieved.

deuterium labelling) was observed in loops or unstructured regions of both the NBD and SBD, whereas regions with highly defined higher-order structures were found to undergo slow HDX (defined as <23% of uptake after 1 min). Specifically, regions of slow HDX were found within α3, 6, and 11 in the NBD, β1 and β2, and αD in the SBD. The inter-domain linker and the SBD C-terminus, which are absent in the crystal structures, underwent very fast exchange, with more than 85% of exchange after 1 min. Interestingly, the β8 and the helix αA of the SBD, despite having defined secondary structure, displayed faster HDX compared to other structured regions, resulting in over 60% of backbone amides exchanged within 1 min of labelling. These regions constitute the interface between the SBDβ and SBDα, which coordinate the binding of the substrate, thus their conformational flexibility may facilitate the key function of the SBD domain.

To reliably identify pH-induced conformational changes in the protein, we corrected our HDX data for the effect of pH on the chemical exchange rate ($k_{ch}$)[48,49] and experimentally validated our approach (Supplementary Note 6 and Supplementary Data 4). Our results show that, while the overall tertiary structure of the NBD and SBD of Hsp70 is retained at pH 5.3, enhanced flexibility and dynamics are observed within both domains. Indeed, at pH 5.3, increased HDX (i.e destabilization) is observed in multiple regions of both SBD and NBD (60% of the whole structure), with the most pronounced effects (more than 15% of the difference in HDX relative to the maximally labelled state) in the SBDβ (aa 430–439), within the binding site for ADP (aa 269–282), at the level of the calcium-binding site (aa 219–241), and in the region comprising residues 85–122. Only the region comprising segment 520–544, within SBD αB, displayed increased protection to HDX at pH 5.3 compared to neutral pH (Fig. 1).

**The point mutation W90F induces a conformational change to the N-terminus of NBD.** Kirkegaard et al. showed that W90F point mutation inhibits the interaction of Hsp70 with the phospholipid BMP. Furthermore, while Hsp70 W90F retained chaperone activity at physiological pH[24], this mutant failed to exert protection toward lysosomal membrane destabilization. To investigate the impact of this mutation on the conformation of Hsp70, we measured the HDX of Hsp70 W90F at physiological and acidic pH and compared these measurements to those of the WT. At pH 5.3, W90F displayed considerable destabilization (i.e. increased HDX) of the NBD, in residues 19–45 and 51–122, and most pronounced in the region spanning amino acids 70–122 (Fig. 2a). The detection of a single peptide comprising residues 87 and 118 did not allow dissecting the HDX for shorter amino acid segments; however, part of the reduction in HDX could be localized to residues 119–122, which are quite remote from residue W90, suggesting that the effect of the mutation is transmitted throughout the sequence 87–122. The major differences in HDX appeared after 10 min, although mutation-induced increases in HDX were also present at early time points (1 min). At physiological pH, after 10 min of HDX, a mutation-induced increase in HDX was confined to the region 87–122 and with a 3.5-fold reduction in the magnitude of the effect compared to that at acidic pH. At 1000 min of labelling, increased HDX was also observed in the regions spanning residues 20–45 and 51–86, although also here the effect resulted less pronounced compared to pH 5.3 (Fig. 2b). By comparing the HDX of Hsp70 W90F at pH 7.4 and 5.3, we saw similar impact of pH on the mutant and the WT, although with some notable differences in the regions impacted by the mutation (Fig. S14). Firstly, unlike for the WT, the region comprising residues 10–17 did not show significant

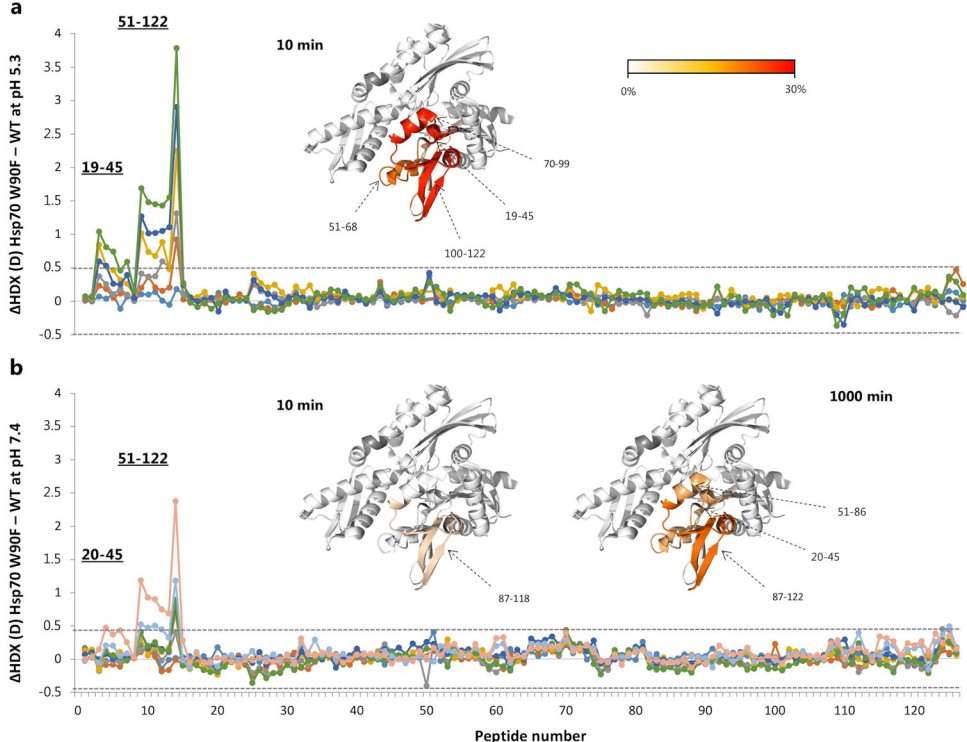

**Fig. 2 Comparison of HDX of His[6]-Hsp70 WT and W90F.** Difference plots illustrating the difference in HDX at pH 5.3 (**a**) and pH 7.4 (**b**) between Hsp70 WT and W90F over the measured time points (cyan line: 5 s, orange line: 15 s, grey line: 1 min, yellow line: 2 min, blue line: 5 min, green line: 10 min, light blue line: 100 min, pink line: 1000 min). Residues comprising a region with a significant difference in HDX are indicated. The peptides are arranged according to their position from N- to C-terminus (Supplementary Data 5, Table 2). Dotted grey lines indicate the 98% CI as a threshold for significance. Regions showing a significant difference in dynamics are coloured in red scale in the crystal structure of the NBD according to their difference in relative fractional uptake normalized to the maximally labelled state. The time point considered for colouring is indicated in the figure. In grey, regions for which no HDX information was retrieved.

differences in HDX. Secondly, a much more pronounced increase in HDX upon lowering pH was observed in the mutant protein in the region encompassing amino acids 19–122 and 134–151. Thus, the mutation not only induces a considerable conformational change in the NBD, but also augments its sensitivity to pH in selected parts.

**Conformational impact of lipid binding on Hsp70 WT.** To evaluate the impact of BMP on the conformational dynamics of Hsp70 in the lysosomes and distinguish its specific binding effect to that of other lipids, we measured the HDX of Hsp70 alone and in the presence of liposomes with or without BMP at pH 5.3. In order to eliminate a potential impact of the histidine tag on the lipid-binding properties of the protein, as indicated in an earlier study[21], the experiment was conducted with the non-tagged form (Hsp70 WT). Furthermore, we did not introduce nucleotides (ATP or ADP) in the reaction mixture, as they impair BMP interaction[23]. We monitored the deuterium incorporation of 125 peptides, spanning 90.9% of the protein sequence (Fig. S15a). Liposomes containing the anionic BMP-induced significant changes in the HDX of Hsp70, both in the NBD and SBD (Fig. 3). In the NBD, decreased HDX was observed in the region spanning the β-strand 2 and 7 and interconnecting segments and helices (residues 19–68), and the region spanning the helices 2–3 and connecting segments (residues 87–118). Two overlapping peptides (69–84 and 69–86) did not show change in HDX in the BMP-bound state, whereas peptides 85–118 manifested a significant decrease in HDX. The peptide comprising residues 70–118, detectable in every analysis, was not detected with sufficient signal-to-noise ratio in the BMP state, which may

represent a signature of strong lipid association even under the acidic quench conditions required by the HDX-MS workflow. In the SBD, liposomes induced reduced HDX in the subdomain β, more specifically between β2 and β6 and interconnecting segments (residues 412–477). Comparing the HDX of Hsp70 when incubated with liposomes with and without the anionic phospholipid (Fig. 3a), we observed that liposomes induced a reduction in HDX of β2 and β3 (residues 412–439) and β4, β5 and β6 (residues 441–477) of the SBD irrespective of whether they contain BMP. Thus, the zwitterionic lipids, particularly phosphatidylcholine, are responsible for this effect. However, the presence of BMP in the liposomes enhances the reduction in HDX observed in β4, β5 and β6 (residues 441–477). The minor reduction in HDX observed in the segment 404–410 could not be unambiguously assigned to the presence of BMP (Fig. 3a). Importantly, no significant changes in HDX were observed in the NBD upon binding of liposomes without BMP, thus the stabilization of this domain is specifically caused by BMP interaction. Taken together, our results clearly demonstrate that, under acidic conditions, the N-terminal portion of the NBD of Hsp70 is the main orchestrator of the association of Hsp70 to the negatively charged phospholipid BMP of the lysosomal membranes.

**Conformational impact of lipid binding on His[6]-Hsp70 WT.** The W90F Hsp70 mutant was engineered with an N-terminal hexahistidine tag. Therefore, in order to exhaustively compare the lipid-binding properties of the WT and mutant form, we first assessed the impact of liposomes on the HDX of an N-terminal hexahistidine-tagged Hsp70 WT (His[6]-Hsp70 WT). The deuterium incorporation of 143 peptides was measured, spanning 92.5%

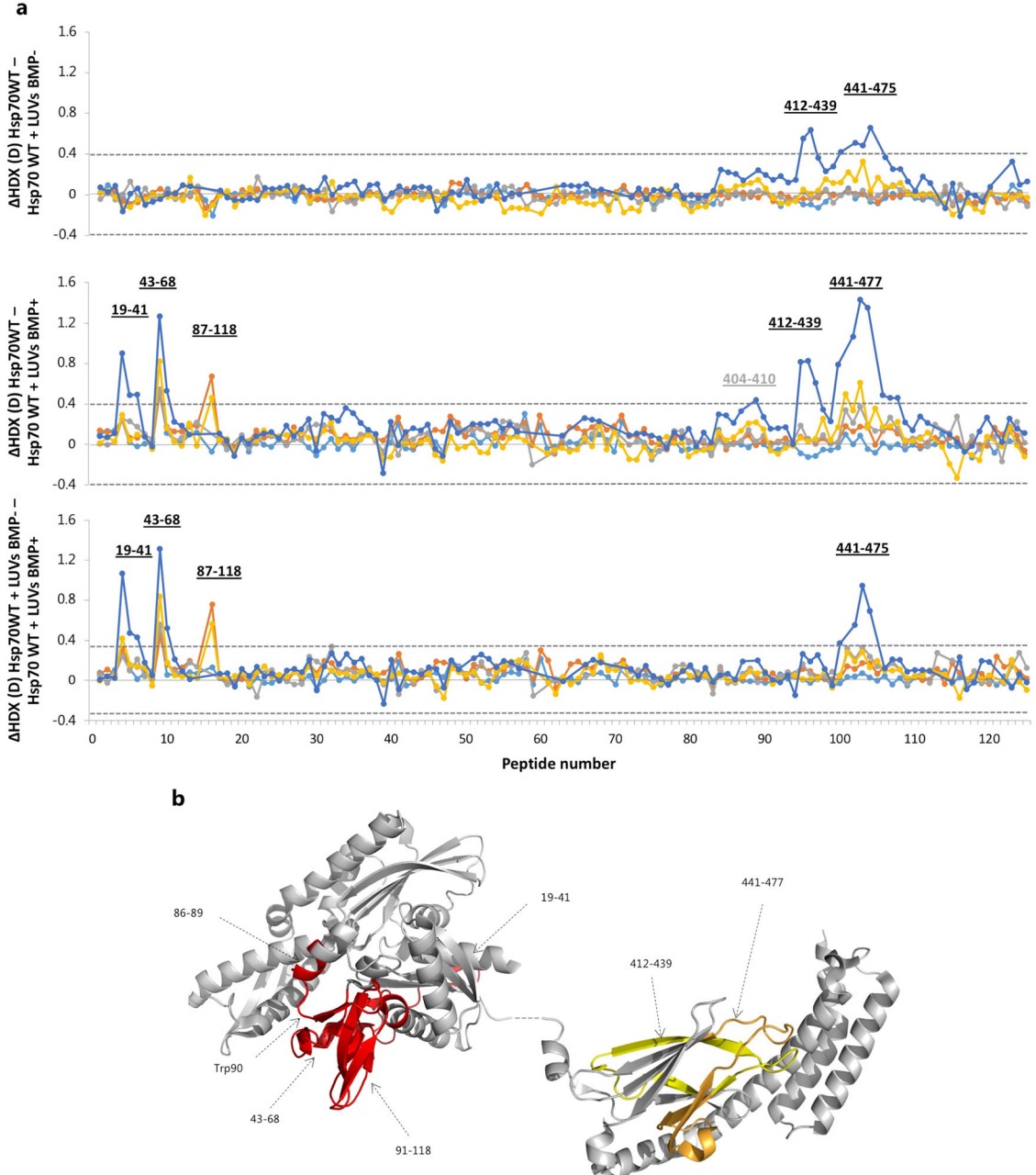

**Fig. 3 Comparison of HDX of Hsp70 WT in the presence of liposomes with and without BMP. a** Difference plots illustrate difference in HDX over the measured time points between Hsp70 alone and Hsp70 incubated with liposomes without BMP (top panel); Hsp70 alone and Hsp70 with liposomes with BMP (middle panel); and Hsp70 with liposomes without and with BMP (bottom panel). Cyan line indicates: 15 s, orange line: 1 min, grey line: 10 min, yellow line: 100 min, blue line: 1000 min. Residues comprising a region with a significant difference in HDX are indicated. The peptides are arranged according to their position from N- to C-terminus (Supplementary Data 5, Table 3). Dotted grey lines indicate the 98% CI as a threshold for significance. **b** Regions showing a significant difference in dynamics are coloured in the crystal structure. Red colour indicates specific binding to BMP, orange indicates unspecific binding to phospholipids enhanced with BMP, and yellow indicates a binding to zwitterionic lipids.

of the protein sequence (Fig. S15b). To note, peptides encompassing the histidine tag could not be identified. Our results (Fig. 4a -top panel- and S16) show that similarities exist between the liposome binding properties of the non-tagged and the His[6]-Hsp70 WT. BMP-specific stabilizing effects were retained between β-strand 5 and 7 and connecting segments (residues 43–68) and the region spanning residues 70–118, as shown for the WT. Notably, none of the overlapping peptides spanning these segments, respectively 45–68/48–68 and 69–86/85–118, presented significant differences in deuterium incorporation,

suggesting a weaker or otherwise different interaction of the tagged form compared to the WT. In conjunction, the lipid-binding properties of non-tagged and His[6]-Hsp70 WT were different in key places in both NBD and SBD. The reduction in HDX in the beta sheets 1 and 2 of the NBD (residues 10–17) of the His[6]-tagged protein (Fig. S16) was not observed in the non-tagged protein. This effect was highlighted irrespective of the presence of BMP in the liposomes and was specifically induced by the presence of the his-tag. In addition, the binding to BMP of the region comprising residues 19–41, observed in the non-tagged

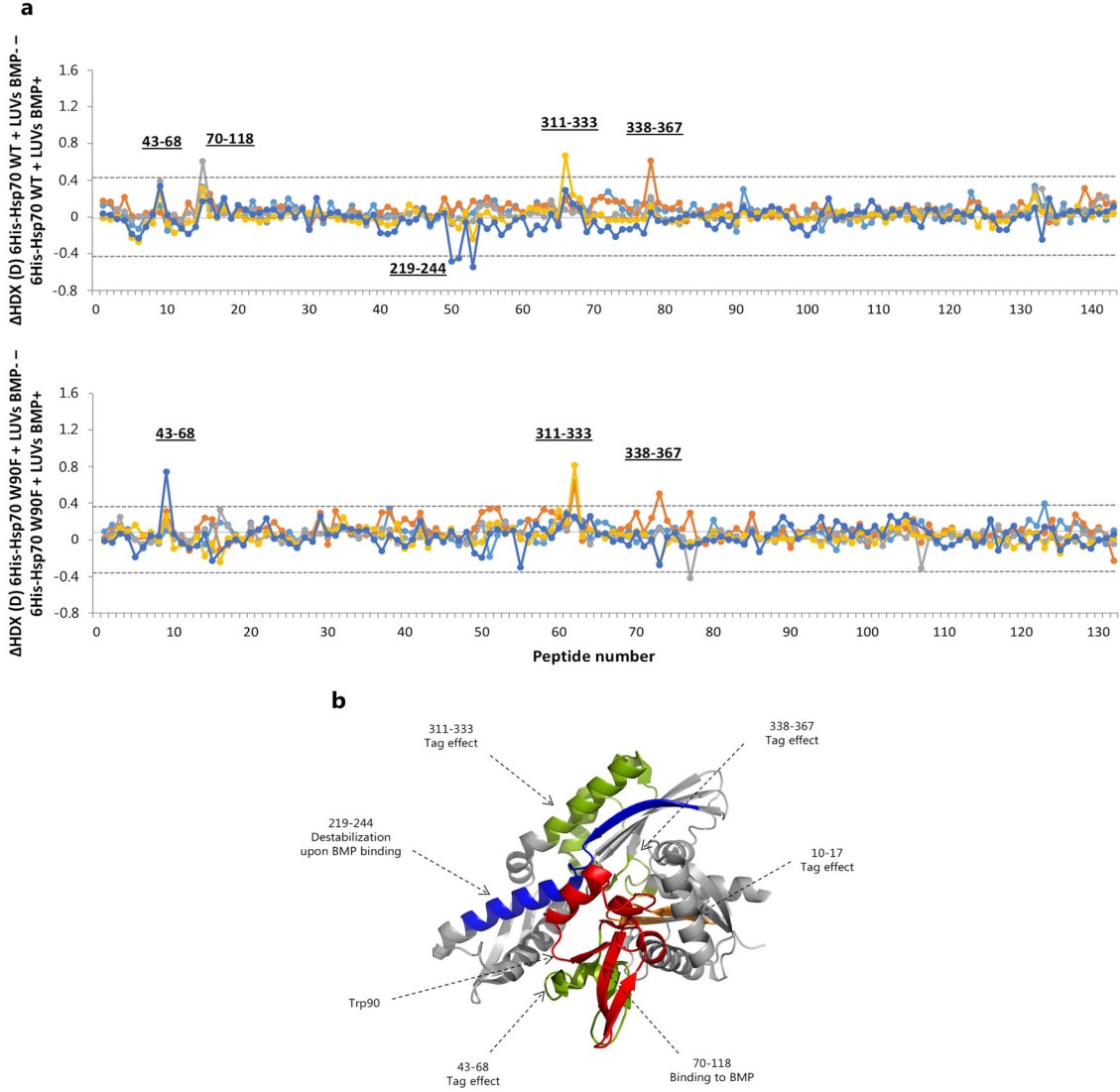

**Fig. 4 Comparison of HDX of the N-terminal hexahistidine tagged proteins (WT and W90F) in the presence liposomes with BMP. a** Difference plots illustrate difference in HDX over the measured time points between His[6]-Hsp70 WT incubated with liposomes without and with BMP (top panel); and His[6]-Hsp70 W90F incubated with liposomes without and with BMP (bottom panel). Cyan line indicates: 15 s, orange line: 1 min, grey line: 10 min, yellow line: 100 min, blue line: 1000 min. Residues comprising a region with a significant difference in HDX are indicated. The peptides are arranged according to their position from N- to C-terminus (Supplementary Data 5, Tables 4 and 5). Dotted grey lines indicate the 98% CI as a threshold for significance. To note, the effect on peptide 43–68 is significant when His[6]-WT is incubated with liposomes with BMP and absent when incubated with liposomes without BMP (Fig. S16), therefore it is attributed to the interaction with BMP lipid. **b** Regions showing significant variation of HDX in His[6]-Hsp70 WT upon BMP binding are deciphered by comparison with the His[6]-Hsp70 W90F analysis and are coloured in the crystal structure of the NBD. Red colour indicates constitutive BMP-specific effect; blue indicates destabilization upon BMP binding; orange indicates unspecific binding to lipids artificially induced by the N-terminal tag; green indicates binding to BMP artificially induced by the tag.

WT, was no longer detectable for His[6]-Hsp70 WT. Additional BMP-specific effects were observed exclusively for the His[6]-Hsp70 WT in the nucleotide-binding pocket of the NBD, within the region spanning residues 311–333 (stabilization) and 219–244 (destabilization), while unspecific binding to lipids was observed within residues 338–367. These effects were not present in non-tagged Hsp70 WT. Finally, no impact of BMP let alone liposomes could be detected in SBD of His[6]-Hsp70 WT, in contrast to the non-tagged Hsp70. As a whole, these data support the key role of the helices 2 and 3 of the NBD (residues 87–118) in the inter-action to the anionic BMP phospholipid, but are also consistent with a drastic alteration of the lipid binding properties of Hsp70 under acidic conditions upon addition of the N-terminal histidine tag[21] (Figs. 4b and S16).

**Conformational impact of lipid binding on His[6]-Hsp70 W90F.** The W90F mutation in Hsp70 has shown to strongly reduce the binding of Hsp70 to BMP, and the mutant chaperone is unable to stabilize the lysosomal membranes[24]. To investigate this further, we measured the HDX of His[6]-Hsp70 W90F in the presence and absence of liposomes (Fig. 4a -bottom panel- and Fig. S17) by monitoring the deuterium uptake of 132 peptides, spanning 92.5% of the protein sequence (Fig. S15c). Strikingly, liposomes with BMP did not induce changes in HDX in the mutant protein in the region encompassing the W90F mutation (peptides 85–118 and 69–118). The same regions displayed major reductions in HDX in the Hsp70 WT proteins (tagged and non-tagged) in the presence of liposomes with BMP (Figs. 3a and 4a -top panel-) and the most pronounced difference in HDX between Hsp70 WT and

W90F mutant (Fig. 2a). Also, the increased HDX observed for the region spanning residues 219–244 in the presence of BMP for the His[6]-Hsp70 WT was not seen for the mutant. In the presence of liposomes without BMP, the HDX of His[6]-Hsp70 W90F was reduced in residues 10–17 (Fig. S17), similarly to what was observed for His[6]-Hsp70 WT, confirming this as a tag-induced effect. Similar to what observed for His[6]-Hsp70 WT, our results showed a decreased HDX for residues 311–333 and 338–367, ascribable to the presence of BMP in the liposomes, effect absent in the non-tagged WT form. We thus extrapolate that also these effects are tag-induced. Analogously, the region spanning residues 43–68 showed stabilization upon BMP association, as detected for the WT constructs (both tagged and non-tagged). We underline that this stabilizing effect followed the tagged construct manner, i.e., no overlapping peptides displaying decreased HDX, hinting at the impact of the tag on this region. Taken together, these data prove that the point mutation decreases the capability of Hsp70 to associate to BMP, but also confirm the unspecific binding effects induced by the N-terminal tag. To summarize, the N-terminal his-tag induces artificial binding to BMP in both WT and mutant forms (regions 43–68 and 311–367), and unspecific lipid interaction on the extreme N-terminal segment of the NBD (region 10–17) (Fig. 4b).

## Discussion

The major human stress-inducible Hsp70 can stabilize endo-lysosomal membranes and revert the cellular pathology associated to lysosomal storage disorders[24]. Therefore, it has been proposed as a drug candidate for Niemann-Pick diseases[24]. Hsp70 binds with high affinity to the anionic phospholipid BMP of the lysosomal membrane, where this lipid plays a crucial role in regulating sphingomyelin metabolism[26,50,51]. Despite several efforts to elucidate the nature of the interaction between the Hsp70 and the lysosomal membranes, we lack a clear understanding of this association both at structural and functional levels. Here we apply HDX-MS to study the conformational dynamics of the full-length Hsp70 at the pH of the cytosol and of the endo-lysosomal lumen, and we map the binding fingerprint, at peptide level resolution, of the anionic BMP and other zwitterionic lipids of the lysosomal membrane. Furthermore, by the analysis of the mutant Hsp70 W90F, which lacks the cytoprotective response[24], we identify regions that could be functionally responsible for the therapeutic potential of Hsp70.

We investigated the dynamics of the chaperone in the cytosol/extracellular environment by performing HDX experiments at neutral pH (=7.4) and without nucleotides. Under these conditions, Hsp70, predominantly in a monomeric form[16], is in a non-BMP-binding conformation[22]. Its conformational properties inferred from HDX (Fig. S12) are in good agreement with the information inferred from the crystal structures of the isolated domains (PDB: 1S3X and 4PO2) and with the full-length bacterial homolog DnaK[14]. As expected, we found the inter-domain linker highly flexible, as the two domains are undocked and functionally independent in the absence of nucleotides[13,52].

Under stress conditions, Hsp70 can translocate into lysosomes, where it binds the inner membranes[31]. Initially, we aimed to study Hsp70 and how it associates with lipids at pH 4.5, mimicking the mature lysosome environment[47]. However, at this pH, we observed that the chaperone unfolds significant parts of its structure in both domains (Supplementary Note 4). We hypothesize that, at pH ≤ 5, the protein transitions into a molten globular state, a fluid conformation that retains secondary structural elements without tertiary structure[53,54]. Conversely, at the lumen pH of late endosomes[47] (pH 5.3), we found that Hsp70 maintains its fold (Supplementary Note 5), albeit both the NBD

and SBD exhibit overall enhanced structural flexibility compared to the cytosolic form (Fig. 1). Notably, the change in pH particularly impacts the sites directly involved in chaperone function, namely the ATP-binding pocket (NBD), the substrate-binding pocket and the lid (SBD), suggesting that Hsp70 incurs a partial loss of chaperone activity in the lysosomes. Intriguingly, the region spanning segment 520–544 within the SBD, which comprises pivotal residues for the ATP-dependent dimerization of the chaperone[16], was the only region found less dynamic at pH 5.3. This may imply a certain degree of dimerization of Hsp70 at the level of the lysosomal membrane, even in the absence of nucleotides. Analogously, also the Hsp70 bacterial homolog DnaK and the mammalian Hsp70 have been reported to dimerize or oligomerize when bound to artificial lipid membranes[55,56]. Importantly, the regions encompassing amino acids 85–122, which includes the most stabilized region upon BMP interaction (see the following discussion), undergoes a significant destabilization when Hsp70 translocates into the acidic lysosomal environment (pH 5.3). The interaction of Hsp70 with BMP is reported weak at neutral conditions ($k_d = 148\,\mu M$)[22] and significantly enhanced at acidic pH[23,24]. To explain the complex formation in the lysosomes, Kirkegaard et al. hypothesized an electrostatic attraction between a positively charged pocket (including W90) in the ATPase domain and the anionic head of BMP[24]. Aligning to this hypothesis, we postulate that H89, whose side-chain pKa is ~6 and is adjacent to W90, acts as pH sensor, as already described for histidine residues[57–60]. Its protonation at lysosomal pH possibly imparts a positive charge to the region and cause a reorganization of the intramolecular electrostatic network, inducing the structural rearrangement into a BMP-binding-competent conformation.

At endo-lysosomal pH (pH 5.3), significant BMP-induced stabilization was observed within the N-terminus of the NBD, localized in the regions spanning residues 19–68 and 87–118 (Fig. 3b). Our results are thus in good agreement with earlier reports indicating the involvement of W90 in the binding to BMP[21,24]. However, at the same time, our data show the involvement of other regions of the ATPase domain (residues 19–68 and 90–118), which have not been implicated in BMP interaction before. Using liposomes without BMP as a "negative" binding control, we distinguished BMP-specific binding effects from those induced by the other main lipid components of the lysosomal membrane (phosphatidylcholine, sphingomyelin and cholesterol). We demonstrate that the NBD binds specifically to BMP, whereas the stabilization observed in the SBDβ (in the region including residues 404–477) mainly arises from the interaction with zwitterionic phospholipids (Fig. 3b), in keeping with previous reports that attribute to the SBD a minor contribution to the BMP association[23]. In addition, our data do not support a transition of the SBD into a molten globule state upon BMP binding at the pH of endo-lysosomes (pH > 5), as earlier proposed[21]. Conversely, from our HDX data, we infer that Hsp70 binds to lipid bilayers in a molten globule conformation at pH of mature lysosomes (pH ≤ 5), as mentioned above.

Importantly, the study published by Kirkegaard et al.[24] was conducted using various constructs of Hsp70 recombinant proteins engineered with an N-terminal hexahistidine tag, not removed after purification. It was later suggested that this modification could alter the lipid-binding properties of the chaperone[21], although no experimental evidences for this were provided. To solve this ambiguity, we aimed to investigate the binding to liposomes of the his-tagged WT form at acidic pH (Fig. 4a). Our findings indicate a direct effect of the His[6]-tag on the lipid-binding properties of the NBD, which in turn indirectly transmits to the SBD. The tagged Hsp70 is still able to associate to BMP in the region spanning amino acids 43–68 and 70–118,

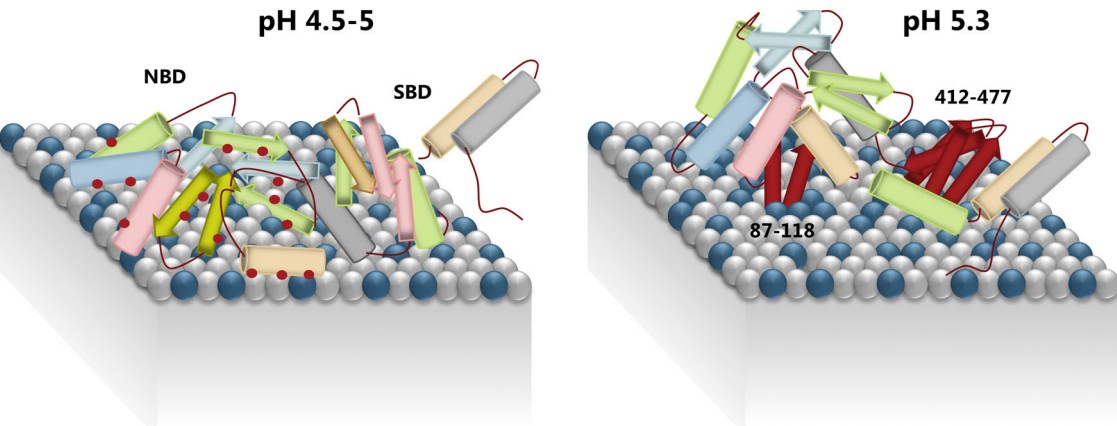

**Fig. 5 Binding model of Hsp70 to the lysosomal membrane at different lumen pH.** The blue spheres represent BMP head, while the grey spheres represent the zwitterionic phospholipids. At pH 5.3 Hsp70 keeps a native-like fold and the NBD becomes immersed in the membrane, whereas the SBD binds peripherally. The elements coloured in red depict the interacting regions. At a pH between 4.5 and 5, Hsp70 adopts a "molten globule" conformation and the NBD interaction to the membrane is mostly mediated by unspecific electrostatic interactions (indicated by red dots) to the anionic BMP.

which includes key residues for the binding, in support to the study of Kirkegaard et al.[24]. Nevertheless, compared to the WT, this interaction appears different. Although the lack of sequence coverage for tag residues prevented to determine whether the tag directly binds the liposomes, we unambiguously observed that the presence of the histidine tag abolishes some conformational effects seen for the WT in the presence of BMP (17–40), while artificially induces others (219–244, 311–333 and 338–367) (Fig. 4b). Contrary to the WT, we also observed no effect of liposomes on the conformational dynamics of the SBD for His[6]-tag Hsp70.

The W90F point mutation strongly reduces the interaction of Hsp70 to liposomes containing BMP[21,24] and abolishes its ability to normalize lysosomes in vivo[24]. We studied the HDX of His[6]-Hsp70 W90F to understand the link between Hsp70 function and conformation, in order to provide a molecular rationale for the lysosome protective response. We observed that the N-terminal his-tagged mutant (Fig. 4a) shows a similar BMP-induced conformational response as the his-tagged WT at the level of 17–40 and 311–333. Also, the covalent modification imparts a weak BMP-binding capacity to the region spanning residues 43–68 to both tagged forms, which overlaps with and modifies the constitutive binding property of this site of Hsp70 WT. More importantly, the mutant form does not manifest a significant stabilization within the region 70–118, which includes W90, when incubated with liposomes containing BMP. Indeed, by comparing the HDX of His[6]-Hsp70 WT and W90F at pH 5.3, we show that the mutation considerably impacts the conformational dynamics of the regions covered by residues 19–45 and 51–122 (Fig. 2). In fact, the substitution of W to F abolishes the hydrogen bonds with Asp69 and Asp86, described in the crystal structure[8] (PDB: 1S3X), likely leading to the significant structural rearrangement of the N-terminus observed at lysosomal pH by HDX. Thus, BMP binding is not solely mediated by an electrostatic interaction between this anionic phospholipid and a region of positive net charge spanning W90, but also requires a binding-competent conformation of Hsp70, which is favoured at low pH. Overall, the considerable impact of the mutation on the NBD at acidic pH explains the abolishment of key interactions of Hsp70 with BMP and the inability of the mutant to correct the dysfunctional lysosomal phenotype[24]. Intriguingly, at neutral pH, only modest differences in HDX were observed between the mutant and WT, which substantiates with its retained chaperone activity under physiological conditions[24]. Because of their critical role in cellular

homeostasis, proteins of Hsp70 family are structurally and functionally conserved in evolution and found in both prokaryotes and eukaryotes[3]. W90 is conserved in the human inducible Hsp70-2 (also called Hsp70-1b) and in the human constitutively expressed Hsc70[61]. Nevertheless, W90 is, for instance, not conserved in the cytosolic Hsp70 isoforms of S. cerevisiae (Ssa1, Ssa2, Ssa3, and Ssa4), where both D69 and D86 are rather conserved. Notably, in Ssa1, Ssa2 and Ssa3, W90 is substituted with a Phe residue[61]. We know that W90F substitution does not abrogate Hsp70 chaperone function[24], however, we observed that this mutation impairs the binding to BMP at lysosomal level. This suggests that human Hsp70 might have specifically evolved to stabilize the lysosomal membrane, a function probably absent in the other organisms, which, indeed, do not possess the same lysosomal machinery as humans. In this regard, both Hsp70-2 and Hsc70, despite the high sequence identity with Hsp70 and the conservation of W90, are not able to stabilize lysosomes[24]. Hsp70-2 shares all but two amino acids (E110D, N499S) with Hsp70[3]. Notably, E110 is comprised within the region identified as primary orchestrator of BMP interaction (87–118) and distant from H89 and W90, in support to a large protein segment participating in the association to BMP and/or in stabilizing the binding-competent conformation.

Based on our HDX-MS analysis, we propose a refined lipid-binding model of Hsp70 in the pH range 4.5–5.3 (Fig. 5). Depending on pH, two states of Hsp70 interacting with the endo-lysosomal membrane can be distinguished. At pH ≤ 5, the NBD and parts of SBD adopts a molten globule conformation, in which the interaction with BMP is mostly mediated by electrostatic forces. At pH 5.3, the region 87–118 of a still-folded Hsp70 establishes a strong direct interaction with BMP, possibly by partially immersing into the lipid bilayer and engaging into hydrophobic interactions, in turn facilitating secondary peripheral associations of other NBD regions (residues 19–68). Furthermore, secondary interactions between SBD (residues 412–477) and the lysosomal membrane are enabled. The primary interaction between region 87–118 and BMP is strong enough that it can occur despite the presence of a His[6]-tag on Hsp70, which otherwise attenuates the binding properties of the NBD and abrogates secondary interactions in the SBD. From a structural point of view, the dual-mode interaction of Hsp70 with lipid bilayers at acidic pH shares some similarities with other protein species that also transition into molten globule states in the lysosomes[62,63]. We note that the Hsp70 protein used in this work

is expressed in bacteria and thus without native human post-translational modifications (PTMs). However, in vivo, proteins of Hsp70 family can be highly modified at post-translational level to regulate localization, refolding activity and interaction with cognate partners[61]. In the human Hsp70, there are several sites found with PTMs in proximity to W90, including phosphorylation of H89[61], which may impact (or regulate) the BMP interaction of this region.

Based on our binding model and earlier works, we conclude that the region 87–118 is also responsible for the lysosomal protective effect of Hsp70, being the unique region that retains the interaction in the wild-type proteins, both active, and fails to bind in the inactive mutant form. It has been demonstrated that the Hsp70-BMP interaction enhances the activity of the lysosomal acid sphingomyelinase (ASM)[24,31], which uses BMP as cofactor[50,51,64–66]. Importantly, an Hsp70-mediated activation of ASM has been proved to induce the lysosome stabilizing effect[24]. We thus hypothesize that this portion of the chaperone (aa 87–118) may help tether together Hsp70, ASM and BMP in a ternary complex, thereby inducing stabilization of the lysosomes. Our findings thus open up for exciting new possibilities of drug design: for instance, a smaller recombinant ΔSBD form, devoid of the regions not involved in BMP-specific interactions, may facilitate Hsp70 targeting of the lysosomes and may constitute a therapeutic option for LSDs. From a method viewpoint, this work also highlights the suitability of HDX-MS to study the peripheral protein-membrane interactions that are important in the complex biochemistry of eukaryotic cellular organelles.

## Materials and methods

**Recombinant Hsp70 proteins (provided by Orphazyme A/S).** Hsp70 WT, His[6]-Hsp70 WT and His[6]-Hsp70 W90F constructs were cloned into the pET-16b vector system (Novagen, Merck KGaA, Darmstadt, Germany). The recombinant Hsp70 protein attached with the hexahistidine tag (sequence GHHHHHHSSGHIEGRH) at the N-terminus were separated using a Ni[2+]-column, with the bound protein eluted with 0.5 mM imidazole at pH 7.4; subsequently, the medium was exchanged for Dulbecco's phosphate-buffered saline (D-PBS) using a PD10 column (Amersham). For Hsp70 WT, the factor Xa recognition site located between the His-tag and the Hsp70 domain was then cleaved by factor Xa and the His-tag and factor Xa removed by filtering through Amicon Ultra 50MWCA (Amersham). The purified proteins were stored at −80 °C.

**Liposome preparation.** Liposomes were prepared from 16:0–18:1 1-palmitoyl-2-oleoyl-glycero-3-phosphocholine (POPC), cholesterol (plant derived), sphingomyelin (from chicken egg yolk), and 18:1S,S bis(monoacylglycero)phosphate (BMP) (Avanti Polar-Lipids Inc., Alabaster, AL, USA). Lipids were dissolved and mixed in chloroform to obtain two types of solutions, containing or not BMP, at the following molar fractions: 0.2 BMP, 0.5 PC, 0.2 cholesterol, 0.1 sphingomyelin (hereinafter referred as BMP+); and 0.7 PC, 0.2 cholesterol, 0.1 sphingomyelin (BMP−). The lipid composition at the selected molar ratios is consistent with Kirkegaard et al.[24] and mimics that of the lysosomal membrane[67]. The solvent was removed by rotary evaporation from the chloroform solutions. The lipid films were stripped with ethanol three times and dried overnight on a rotary evaporator to remove trace amounts of chloroform. The lipid films were then dispersed in the aqueous (undeuterated) acetate buffers (pH 4.5, 5, and 5.3) to give a total lipid concentration at 3.8 mM, and the suspensions were sonicated for 5 min at 25 °C to minimize lipid loss on the glass walls of the flasks. Subsequently, the suspensions underwent six cycles of vigorous vortexing (2 min) and sitting in a water bath at 47 °C (10 min), followed by 1 h-rest at room temperature for annealing. The resulting multilamellar vesicles were extruded ten times through a 100 nm pore size polycarbonate membrane filter with a LIPEX® Thermobarrel Extruder (Transferra Nanoscience Inc.) held at 47 °C. The mean diameter and dispersion of the two liposome suspensions was measured in triplicates by dynamic light scattering on a Zetasizer Nano ZS (Malvern Panalytical) (Supplementary Note 2). Liposomes were stored at 4 °C until usage for deuterium labeling.

**Hydrogen-deuterium exchange.** The low-pH HDX buffers (pH 4.5, 5 or 5.5) contained 20 mM sodium acetate, while the neutral-pH HDX buffer (pH 7.4) contained 20 mM phosphate. The undeuterated and 99%-deuterated buffers were prepared by a 100-fold dilution of a 2 M aqueous stock solution in MilliQ water or heavy water respectively and added with an amount of NaCl to yield a final ionic strength of 154 mM. Measurements of pH were performed with an H2O-calibrated 827 pH lab pH-meter (Metrohm) and buffer compositions were optimized by

considering suitable for this study undeuterated buffers with a pH within ±0.050 units of the aimed one (Supplementary Note 1). The pD was calculated according to the Eq. (1):

$$pD = pH_{read} + D_{frac} \times 0.4, \qquad (1)$$

where $pH_{read}$ is the directly read value of the deuterated buffer returned by the H2O-calibrated pH meter, $D_{frac}$ is the fraction of deuterium in the HDX labelling buffer, and 0.4 is the value of the glass electrode correction factor[68].

**Correction of HDX data collected at different pH values.** In order to compare the HDX of Hsp70 under the two different pH conditions (7.4 and 5.3), we corrected for the effect of pH on the chemical exchange rate ($k_{ch}$). Near or above neutral conditions (pH 5–10), the $k_{ch}$ is primarily base-catalyzed and can be described with Eq. (2):

$$k_{ch} \sim k_{OH}[OH^-] = A \exp\left(\frac{-E_a}{RT}\right)[OH^-], \qquad (2)$$

where A is the frequency factor, and $E_a$ is the activation energy of the dominating base-catalyzed amide hydrogen exchange reaction. At constant temperature and ionic strength, to calculate the ratio of amide hydrogen exchange rate constants at two pH conditions, Eq. (2) evolves into Eq. (3):

$$\frac{k_{ch1}}{k_{ch2}} = \frac{[OH^-]_1}{[OH^-]_2} = \frac{k_w/[H^+]_1}{k_w/[H^+]_2} = \frac{10^{-pH_2}}{10^{-pH_1}} = 10^{pH_1 - pH_2}, \qquad (3)$$

The pH effect on the $k_{ch}$ was thus corrected by multiplying the observed HDX times at pH 7.4 by a factor equal to $10^{pH_{7.4} - pH_{5.3}}$. Considering the values of the $pH_{read}$ of the deuterated buffers, respectively 7.36 and 5.38, a correction factor of 96.38 was applied to convert the HDX times at pH 7.4 into HDX times at pH 5.3. The suitability of the calculated correction factor was also experimentally validated (Supplementary Note 6).

**Continuous labelling experiment of His[6]-Hsp70 WT and W90F at pH 5.3 and 7.4.** His[6]-Hsp70 WT and W90F were diluted into undeuterated low- or neutral-pH buffers and the solutions were equilibrated for 20 min. HDX was initiated by a 10-fold dilution into the corresponding deuterated HDX buffer at room temperature resulting in 89% deuterium content in the reaction mixture. At various time intervals, an aliquot was withdrawn and quenched 1:1 with ice-cold phosphate 300 mM phosphate buffer with 6 M Urea (final pH_read 2.3). After quenching, samples were immediately frozen and kept at −80 °C until LC-MS analysis. The time points referenced for the experiment at pH 7.4 were 5 s, 15 s, 1 min, 5 min, 10 min, 100 min and 1000 min, which yield the following pH-corrected time intervals (see earlier) at pH 5.3: 8 min 2 s (5 s*), 24 min 6 s (15 s*), 96 min 23 s (1 min*), 192 min 46 s (2 min*), 481 min 55 s (5 min*) and 963 min 50 s (10 min*). The values in brackets with (*) indicate the corresponding time period at physiological pH. Given the large conversion factor, the longest deuteration at pH 5.3 was restricted to 963 min 50 s (10 min as converted on-exchange time at pH 7.4), because of the potential vulnerability of the protein to long-term storage (>1000 min) under acidic conditions. The labelling for 15 s and 1 min was performed two additional times, producing three technical replicates for those time points. Maximally labelled controls were also included in the experiment.

**Continuous labelling experiment to assess lipid binding at pH 5.3.** The three Hsp70 variants (Hsp70 WT, His[6]-Hsp70 WT, and His[6]-Hsp70 W90F) were incubated with freshly made LUVs BMP+ and BMP− (protein:lipid ratio 1:1000) or with the undeuterated low-pH HDX buffer for 20 min, in order to allow binding to equilibrate prior to the labelling. HDX was initiated by a 4.6-fold dilution into the acetic deuterated buffer (pH 5.3) at room temperature resulting in 78% deuterium content in the reaction mixture. At various time intervals (15 s, 1 min, 10 min, 100 min, and 1000 min), an aliquot was withdrawn and quenched 1:1 with ice-cold phosphate buffer with 6 M Urea (final pH_read 2.3). After quenching, samples were immediately frozen and kept at −80 °C until LC-MS analysis. The deuteration for 10 and 100 min was performed two additional times, producing three technical replicates for those time points when labelling the hexahistidine-tagged proteins. For Hsp70 WT, replicates were produced at time points of 15 s, 1 min, and 100 min. Maximally labelled controls were also included in the experiments.

**Maximally labelled controls.** The proteins were diluted in the aqueous buffer and the exchange reaction was initiated by dilution with fully deuterated 6 M GndDCl in 20 mM Tris buffer (pH 5) at room temperature, to allow maximal deuteration of the unfolded protein. The deuterium content in the reaction mixture was identical to the corresponding HDX experiment. After 24 hours, the reaction was quenched with 1:1 dilution into ice-cold 300 mM phosphate buffer (final pH_read 2.3) and samples were kept frozen at −80 °C until LC-MS analysis.

**LC-MS analysis.** Peptide identification was performed by tandem mass spectrometry on the Hsp70 WT and His6-Hsp70 W90F non-deuterated proteins. Samples were injected into a refrigerated UPLC system (NanoAcquity, Waters) with all

chromatographic elements held at 0 °C. The protein passed through an in-house packed column containing pepsin immobilized on agarose resin beads at 20 °C. The generated peptides were trapped on a Vanguard column (BEH C18, 130 Å, 1.7 µm, 2.1 mm × 5 mm; Waters) and desalted for 3 min in solvent A (0.23% formic acid in MilliQ water, pH 2.5) at 200 µl/min. Subsequently, the peptides were separated over an Acquity UPLC column (C18, 130 Å, 1.7 µm, 1.0 mm × 100 mm; Waters) protected by a Vanguard column, with a 9 min-linear gradient rising from 8 to 50% of solvent B (0.23% formic acid in acetonitrile) at 40 µl/min. Peptide ions were fragmented in a hybrid ESI-Q-TOF mass spectrometer (Synapt G2-Si, Waters) by collision-induced dissociation using argon as collision gas and MS/MS spectra were acquired either in data-dependent (DDA) or data-independent acquisition (DIA) mode. The acquired MS/MS data were processed with ProteinLynx Global Server (PLGS) version 3.0 (Waters). Peptides identified by DDA had to present a mass error below 15 ppm for the precursor ion, along with passing manual inspection of the fragment spectrum in PLGS. DynamX version 3.0 (Waters) was used to filter peptides identified by DIA by selecting those fragmented at least 2 times, presenting 0.2 fragments per amino acid, and showing mass error for the precursor ion below 10 ppm. In addition, the peptides had to be identified in 3 out of 4 of the acquired MS/MS files. Selection of the quench buffer ensuring maximal peptide signal intensities was performed as described in SI (Supplementary Note 3). For deuterium labelled proteins, frozen quenched samples were thawed in a table top centrifuge and subjected to LC analysis identical to the one performed for undeuterated samples. Following the chromatographic separation, the peptides were analyzed setting the MS in positive ionization mode with spray voltage of 2.8 kV, and the ions were further separated by ion mobility for enhanced peak capacity. The MS spectra were acquired in range from 50 to 2000 $m/z$, performing a scan every 0.5 s. Human Glu-fibrinopeptide (Sigma-Aldrich, St. Louis, MO) served as a continuous (lock-spray) calibration standard and the peptide signal was recorded throughout all the analysis every 30 s.

**Statistics and reproducibility**. DynamX 3.0 was used to calculate the deuterium contents of peptides generated from the labelled proteins. The threshold for a statistically significant difference in HDX between two experimental states was established based on an approach described earlier[69]. Briefly, a confidence interval (CI) was calculated based on the standard deviation (SD) of peptide deuterium contents for time points performed in triplicates. For each state, the SDs were averaged using the root-mean-square, according to Eq. (4):

$$SD_{state} = \sqrt{\frac{\sum SD_i^{\,2}}{N}}, \qquad (4)$$

where $N$ is the number of peptides considered, multiplied by the number of time points performed in triplicate. A pooled SD for the difference between two states, state A and B, was subsequently calculated using Eq. (5):

$$SD_{pool} = \sqrt{SD_{stateA}^{\,2} + SD_{stateB}^{\,2}}, \qquad (5)$$

The pooled SD was used to calculate the CI at the significance level of 98% with a zero-centered average difference in deuterium content, considering a two-tailed distribution with two degrees of freedom ($n = 3$), by using Eq. (6):

$$CI = \mp 6.965 \times \frac{SD_{pool}}{\sqrt{n}}, \qquad (6)$$

The deuterium content of every peptide generated from the maximally labelled controls served to calculate the back exchange (BE), by the following equation:

$$BE\% = \left(1 - \frac{N_{max}}{N \times D_{frac}}\right) \times 100, \qquad (7)$$

where $N_{max}$ is the number of exchanged amides (i.e. deuterium content) of the maximally labelled peptide, $N$ is the number of exchangeable backbone amides of the peptide (calculated as number of amino acids – number of prolines − 1), and $D_{frac}$ is the fraction of deuterium in the HDX labelling buffer. Calculated BE values were used to validate LC-MS system performance and to support the HDX-MS analysis.

**Reporting summary**. Further information on research design is available in the Nature Research Reporting Summary linked to this article.

## Data availability
Data supporting the findings of this manuscript are available from the corresponding author upon reasonable request. According to the community-based recommendations[39], to allow access to the HDX data of this study, the HDX summary table and the HDX data tables are included in Supplementary Data 5 (Tables 1–5). All the deuterium uptake plots of the experiments presented in this work are available on the figshare data repository at https://figshare.com/s/a4940a2f06a34e969d7c.

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

## Acknowledgements

The authors gratefully acknowledge funding from the Danish Council for Independent Research (Sapere Aude grant DFF-4184-00537A to K.D.R.). We also acknowledge technical support provided by Fabrice Rose for training Calvaresi V. in liposome preparation.

## Author contributions

K.D.R. and T.K. conceived the study. V.C., K.D.R., N.H.T.P. and T.K. designed the experiments. T.K and N.H.T.P. provided the recombinant proteins. V.C., L.T.T. and S.B.L. performed the experiments. V.C., L.T.T., S.B.L. and K.D.R. analyzed the data. V.C. and K.D.R. drafted the manuscript and all authors commented on the final manuscript.

## Competing interests

T.K. and N.H.T.P. are shareholders and employees of Orphazyme A/S. V.C., L.T.T., S.B.L. and K.D.R. have no competing interest to disclose.
