## [Transparent Peer Review File · Communications Biology]

Reviewers' comments:

Reviewer #1 (Remarks to the Author):

This research manuscript describes interesting mechanistic details of the binding of HSPA1A to a particular lipid. The manuscript uses a relatively new technique and concept to identify the conformational changes of the protein caused by different pH and liposomes composed of different lipids. The manuscript is well-presented, the results well-defended, and the conclusions appropriately presented.

Below I provide some comments and questions to explain the presented concepts. Please note that they are presented in order of appearance in the text and their order does not represent their importance.

1. Page: 2

Line 47

Hsp70 embedding: Based on the public literature whether the protein embeds or binds peripherally depends on the saturation level of the lipids. Therefore, although HSPA1A binds to BMP peripherally in the cited literature, these studies did not test the saturated forms of the BMP; thus, whether it embeds might not be resolved yet. Maybe the author could clarify this issue.

2. Page: 4

a. Line 83 (and in several other parts, please see below): I am kind of confused about the validity of the sentence concerning phosphatidylcholine's charge. Although the choline is positively charged the total charge of PC is 0. Thus, PC is neutral in the pH range used.

b. Line 94: it would be great to explain this particular part of the text for the technique and the data it provides. I think that a short explanation on how the method can provide structure information would make the concept clearer (e.g., provide information of structured vs unstructured regions?)

3. Page: 5

Line 118: please change letter to latter

4. Page: 6

Line 126 (and figure 1)

If I am not interpreting the results erroneously, a shift of the pH alone causes major conformational changes in similar or the same regions that BMP binds; what does that mean in respect to the binding? A little more in-depth explanation would be great.

5. Page: 7

Line 143

I hope I did not miss this information in the cited paper but I cannot seem to find the experiments proving the intact chaperone function of the mutant; I can only find a reference to a "data not shown: for the ATP hydrolysis".

6. Page: 8

Line 160:

I apologize if I am not interpreting the data properly, but my interpretation of this is that there is a difference irrespective of the lipids in this region. A clearer explanation would be appreciated.

7. Page: 9

a. Line 176

This is a major question that I have concerning the W90 position. Is the W90 responsible for the binding to BMP or its mutation alters the protein conformation and this results in loss of binding? As it is written here it is not clear; a clarification, here and in other places in the text, would be great.

b. Line 191:

"sufficient intensity": please explain this a little bit more. It is unclear why this is the case.

c. Line 198: please see comment 2a about PC being positive.

8. Page: 13

a. Line 265

(Please see also comment 7a). The sentence reads as if the mutation causes lipid-binding loss but the results provided describe protein alterations that affect BMP binding. Please clarify and aim to be consistent.

b. Line 266

I understand that the site with the poly-His tag could not be mapped, but I am wondering how sure are we that this is binding? I am not arguing that at that pH the poly-His do not affect binding, I am just trying to understand it. Could these alterations be the result of the tag being so close to the binding site (and the region that changes by the pH shift) that causes these discrepancies?

Furthermore, the way the text is written, it implies that any tag (even a C-terminal) one would have the same effects. Is this true? If this is not, or it is not clear then maybe in addition to clarifying the above issues, it would be important to note everywhere in the text that this is an N-tagged protein.

c. I am wondering what could explain the large differences in DeltaHDX values between the experiments presented in the different figures. What I am asking is: if we run this experimental setup and repeat it with different batches of protein will we get the same results? Will we see the same picks and the same levels of change or not? A clarification would be great here.

9. Page: 14

a. Line 301:

I am not sure what this sentence means because the region that binds or does not bind was known. Is there something that I missed?

b. Line 306

Fig. S12: I am not sure how I can see the conformational properties in this figure. Please clarify if possible.

10. Page: 15

a. Line 317

It is not clear from the cited literature that HspA1A binds to the lysosomal membrane. Please clarify.

b. Line 325

Chaperone function alteration by pH: I am not sure I understand this finding. I am not arguing it is not true, just where does that come from?

c. Line 331

HspA1A dimerization: Could this be supported by other lipid membranes in which Hsp70 was found to be a dimer or oligomer (sulfatide or phosphatidylserine)?

11. Page: 16

a. Line 338

The idea of a lipid-binding pocket formation by pH change is great, but there is still binding at pH 7, albeit binding at lower pH is higher but still there is considerable binding at neutral pH. Could it be enhanced at these pH values because of the more pronounced conformational changes? Although H89 seems to be important, holistically the results do not seem to point to a residue but to a major conformational change of a region. What about other amino acids that are already protonated within this pH range (lysine or arginine)?

b. Line 346 (please also refer to comments 7a and 13a)

Is W90 responsible for binding or not?

c. Line 353 (please also refer to comments 2a and 7c)

PC charge

12. Page: 17

a. Line 363 (please also refer to comment 8b)

How sure are we that all poly-His tags will have the same effect? Maybe be specific about the N-terminal tag?

b. Line 373

Could this be because the flexible 6H tag is so close structurally to the "binding site"?

13. Page: 18

a. Line 392 (please also refer to comments 7a, 13a, and 16b)

"The mutation impacts the dynamics of the regions". If I am interpreting this sentence correctly then it might be changing the protein and not the binding, which also changes by the pH; is this correct?

Please clarify.

b. Line 393

"Hydrogen bonds D69 and D86 with W90". I am not sure that I could find this information in the cited paper. Please clarify.

14. Page: 19

Line 419

Figure 5. This is a great figure, but I was hoping to make it more informative by mapping the regions and adding the information of the N- or C- termini.

15. Page: 20

a. Line 443

Which clone did the authors use, which accession number?

b. Line 453

Which lipids used specifically? Source, company.

c. Line 456

Would it be possible to provide the rationale of using the specific lipid mixtures? Why these specific lipids and why in these molar ratios?

16. Page: 21

Line 464

Would it be possible to provide the rationale of using 47C?

Reviewer #2 (Remarks to the Author):

The molecular chaperone Hsp70 is known to play a role in stabilizing the endo-lysosomal membranes. In this study, the authors follow up on their previous seminal studies in exploring the nature of interaction between the Hsp70 and lysosomal membranes. Hydrogen-deuterium exchange (HDX) MS is used to understand conformational changes of Hsp70 under different pH environments. Specific regions of Hsp70 that interact strongly with BMP are identified and the impacts of both the W90F mutant as well as HIS tag are explored. Overall this study is rigorous and well-written, and contributes greatly to our understanding of the protective effects of Hsp70 in lysosomal storage disease. Specific comments:

1. Line 96-98, please provide a reference for the cut-off values for fast and slow HDX.
2. Line 118: Spelling mistake, should be "latter"
3. Fig. 1: The graph axis shows a pH 5.3, but description mentions pH 5.4.
4. Fig. 1: Please increase the font size on the x-axis "peptide number"
5. For all graphs, please use "." Instead of "," for numerical values
6. Line 191-192: Mentions that residues 70-118, detectable in every analysis was not detected with sufficient intensity in the BMP state, which may represent a signature of strong lipid association. However, In the figure 3A (middle and lower panel) it does show some increase in HDX with BMP. Please could you comment on this?
7. All the work presented is in vitro. It would be nice if the authors could comment on how well their findings may translate to a cellular context. For example, do the authors predict that co-chaperones of Hsp70 like DNAJA1 are involved in lipid binding given that they bind to Hsp70 near the proposed lipid binding region? In addition, Hsp70 is highly modified at the post-translational level (the chaperone code). We ask the authors to add the appropriate reference (Pubmed: 32518165) and discuss briefly the potential impact of PTMs on lipid binding. For example, there are several phosphorylated and acetylated amino acids in close proximity to W90 that may alter lipid binding. Finally, we would like the authors thoughts on why W90 is not conserved in other organisms (for example yeast Ssa1).

Reviewer #3 (Remarks to the Author):

In the manuscript by Calvaresi et al. more structural information is provided about the interaction of

the heat shock protein 70 (hsp70) to the anionic bis-(monoacylglycerol)-phosphate (BMP) usually present in the membranes of intraluminal vesicles in lysosomes. The authors build up on earlier studies where this binding was defined to occur within the nucleotide binding domain (NBD) of Hsp70 and a Hsp70 W90F mutant was reported to have almost lost interaction to BMP (e.g. ref#24). The current study makes use of hydrogen-deuterium exchange mass spectrometry analysis. The knowledge about the importance of the NBD and the relevance of the Hsp70W90F mutant for mediating the BMP binding is confirmed. A lipid binding dual mode model is suggested which apparently is pH dependent, i.e. dependent on the subcellular localization (late endosomes vs lysosomes). At pH5.3 Hsp70 exerts a native fold and NBD binds to the membrane and the substrate binding domain is more peripherally localized. At a more lysosomal pH of 4.5 the authors suggest a globule conformation and interactions to the BMP-containing membrane mediated by unspecific electrostatic interactions.

Although the manuscript is clearly written I doubt that the new structural insights reported here go far beyond the knowledge which was already available. It remains a bit vague in how far the hydrogen-deuterium exchange mass spectrometry data can be translated in a more physiological set up, e.g. by addressing selective Hsp70 mutations and analyzing lysosomal functions under different conditions. At the present stage this study seems more appropriate for a specialized journal. I would have also expected -at least for this journal- experiments (structure-function analyses, CRISPR knockin cell lines) which address the suggested model in suitable cell based assays (e.g. in NPC1 KO cells).

Additional comments:

More (experimental and literature) information is required how and under which conditions a cytosolic protein (hsp70) may enter the lumen of endosomes and lysosomes. How is the translocation of hsp70 from the cytosol to endocytic vesicles mediated and what is the fate of hsp70 in such an enigmatic translocation process?

Is the BMP found at the inner side of the limiting membrane of lysosomes or as reported more abundantly within intraluminal vesicles of late endosomes? To which membrane subdomain does hsp70 bind to and what is the dynamics of such a binding process? How can hsp70 overcome the glycocalyx barrier which is present at the inner membrane of lysosomes. Experiments performed in liposomes do not necessarily represent a very physiological situation.

Can the critical role of the region between hsp70 aa87-118 more precisely defined and can selected mutants be tested in a suitable set up to restore lysosomal storage (in NPC-deficient cells)? More specifically: the critical H89 residue which is adjacent to the W90 residue seems to trigger the structural rearrangement into a BMB binding-competent conformation. How can this be explained? Further validation by cell-based assays are required. A possible dimerization of Hsp70 is discussed but not experimentally addressed. Dimerization may well affect binding to BMP and requires more attention

The authors mention (line 76) a BMB interaction in the cytosol. Where should this occur?

Reviewer: 1

This research manuscript describes interesting mechanistic details of the binding of HSPA1A to a particular lipid. The manuscript uses a relatively new technique and concept to identify the conformational changes of the protein caused by different pH and liposomes composed of different lipids. The manuscript is well-presented, the results well-defended, and the conclusions appropriately presented. Below I provide some comments and questions to explain the presented concepts. Please note that they are presented in order of appearance in the text and their order does not represent their importance.

1. Page: 2

Line 47

Hsp70 embedding: Based on the public literature whether the protein embeds or binds peripherally depends on the saturation level of the lipids. Therefore, although HSPA1A binds to BMP peripherally in the cited literature, these studies did not test the saturated forms of the BMP; thus, whether it embeds might not be resolved yet. Maybe the author could clarify this issue.

Our response:

We thank the reviewer for pointing this out and we think the reviewer's remark is correct. We modified the sentence, which now reads as "The binding of Hsp70 to BMP has been studied in the context of a lipid bilayer containing phospholipids with monounsaturated fatty acid chains (18:1 Δ^9). The chaperone does not integrate into such lipid bilayers containing BMP, rather associates peripherally^{21,24}." (lines 47-50 with changes not shown).

2. Page: 4

a. Line 83 (and in several other parts, please see below): I am kind of confused about the validity of the sentence concerning phosphatidylcholine's charge. Although the choline is positively charged the total charge of PC is 0. Thus, PC is neutral in the pH range used.

Our response:

We thank the reviewer for giving us the opportunity to clarify that. Indeed, PC and sphingomyelin are zwitterionic lipids, with the head group carrying both a positive (amine group) and a negative charge (phosphate group). We indicated them as "positively charged" to more clearly differentiate them from BMP, which only carries the negative charge, but we recognize that this definition is not totally correct. We have now changed "positively charged" in "zwitterionic" throughout the manuscript.

b. Line 94: it would be great to explain this particular part of the text for the technique and the data it provides. I think that a short explanation on how the method can provide structure information would make the concept clearer (e.g., provide information of structured vs unstructured regions?).

Our response:

We thank the reviewer for giving us the opportunity to elaborate. In the interest of limiting the length of the manuscript, we had skipped this in the submitted version. We have now added this additional text to the part of the introduction where the HDX-MS is first introduced (lines 66-69 with changes shown):

"Indeed, high % deuteration after a brief exposure time indicates that a given region has high flexibility in solution. This is generally seen for loops and regions of random coil structure.

Conversely, folded regions generally show a significantly low % deuteration due to the presence of stable hydrogen-bonding networks”.

Furthermore, we have added in the results session that the percentage of HDX were normalized with those of the maximally labelled state, i.e. an unfolded fully deuterated form of Hsp70 (lines 93-95 with changes not shown):

“Generally, fast HDX (defined here as more than 80% of uptake relative to the maximally labelled state after 1 min of deuterium labelling)”.

3. Page: 5

Line 118: please change letter to latter.

Our response:

Thank you for pointing out the typo. We have made the correction.

4. Page: 6

Line 126 (and figure 1). If I am not interpreting the results erroneously, a shift of the pH alone causes major conformational changes in similar or the same regions that BMP binds; what does that mean in respect to the binding? A little more in-depth explanation would be great.

Our response:

The reviewer is correct in pointing this out. Indeed, in lines 321-324 (with changes not shown) of the discussion we have highlighted that “Importantly, the regions encompassing amino acids 85-122, which includes the most stabilized region upon BMP interaction (see following discussion), undergoes a significant destabilization when Hsp70 translocates into the acidic lysosomal environment (pH 5.3).”. Briefly, indeed, we believe that the pH-induced changes in Hsp70 in those particular regions explain the increased binding affinity of Hsp70 to BMP at lysosomal acidic pH relative to at physiological pH ($k_d=148 \mu\text{M}$). It has previously been demonstrated that the binding has a strong electrostatic component. However, given the significant difference in conformational dynamics of the protein at acidic pH in region 85-122 that we observed, we think that the protein also adopts a more binding-competent conformation at acidic pH, which is absent at neutral pH. As we postulate on lines 328-329, this structural rearrangement might be triggered by His89 getting protonated at acidic pH, changing the intra-molecular electrostatic interaction, thus the protein conformation. To better explain that, we have revised this part in the discussion, which now reads “...we postulate that H89, whose side-chain pKa is ~ 6 and is adjacent to W90, acts as pH sensor, as already described for histidine residues⁵⁶⁻⁵⁹. Its protonation at lysosomal pH possibly imparts a positive charge to the region and cause a reorganization of the intramolecular electrostatic network, inducing the structural rearrangement into a BMP binding-competent conformation.

” (lines 328-332, with changes not shown).

5. Page: 7

Line 143. I hope I did not miss this information in the cited paper but I cannot seem to find the experiments proving the intact chaperone function of the mutant; I can only find a reference to a “data not shown: for the ATP hydrolysis”. The cited paper states that the mutant W90F retains the functional aspects of the chaperone activity based on measurements of the ATP hydrolysis and the folding of the luciferase.

Our response:

The reviewer is correct in pointing out that these data are not shown in the cited paper (Kirkegaard et al., 2010), which we believe to be common practice. Our HDX-MS analysis reveals that the mutation (at neutral pH, Fig. 2, bottom panel) has no impact on the regions involved in coordinating the ATP or the unfolded substrate, and generally very limited conformational impact compared to that at acidic pH. This aligns with the data from Kirkegaard et al. (2010).

6. Page: 8

Line 160: I apologize if I am not interpreting the data properly, but my interpretation of this is that there is a difference irrespective of the lipids in this region. A clearer explanation would be appreciated.

Our response:

We are not sure to fully understand the question. If our interpretation of the question is correct, the reviewer is pointing out the regions showing a difference in dynamics between the WT and mutant form are also the regions involved in the binding to BMP. This is correct and this aligns with our findings that the mutant W90F does not show a difference in HDX in region 70-118 when the protein is incubated with liposomes containing BMP (shown in fig. 4A and discussed in lines 369-371 with changes not shown). To meet the reviewer's request of clarity, we have now added "and the most pronounced difference in HDX between Hsp70 WT and W90F mutant (Fig. 2) (line 251-252 with changes not shown) in the result session "Conformational impact of lipid binding on His6-Hsp70 W90F", where we think this explanation could better fit.

7. Page: 9

a. Line 176. This is a major question that I have concerning the W90 position. Is the W90 responsible for the binding to BMP or its mutation alters the protein conformation and this results in loss of binding? As it is written here it is not clear; a clarification, here and in other places in the text, would be great.

Our response:

We thank again the reviewer for giving us the opportunity to clarify. This is really an important point that, we admit, should be better addressed in the discussion. To unambiguously answer the reviewer's question, we should have a spatial resolution at single amino acid level for W90, a resolution that, at the current state-of-art, HDX-MS experiments cannot achieve along the whole protein sequence. Alternatively, we should have a peptide spanning W90 showing no difference in HDX to exclude that W90 is directly involved in the binding. With the data in our hands, we can answer the reviewer's question in the following manner:

1. The point mutation causes an alteration in protein dynamics in region 19-45 and 51-122, and most pronounced in the region spanning amino acids 70-122 (Fig. 2, top panel). This is presumably the reason why the binding in region 43-68 is retained also in the mutant, while the binding in region 87-118 is abrogated.
2. For the reasons elucidated above, we cannot unambiguously determine whether W90 binds to BMP or not, even though this seems likely to us, as it is widely accepted that Trp residues play a crucial role

as membrane anchors. In the improbable scenario where the binding site excludes W90, this would be anyway very close to this residue.

3. We can certainly say from our data that the W90F mutation alters Hsp70 conformation in several regions and that this results in the loss of lipid binding in the region 87-118. From a physicochemical point of view, we consider it unlikely that a single point mutation, without a substantial structural change, could totally abrogate binding and the large difference in HDX observed upon mutation better supports a loss of binding competent conformation

We have now added the following paragraph to the discussion (lines 373-380 with changes not shown):

“In fact, the substitution of W to F abolishes the hydrogen bonds with Asp69 and Asp86, described in the crystal structure⁸ (PDB: 1S3X), likely leading to the significant structural rearrangement of the N-terminus observed at lysosomal pH by HDX. Thus, BMP binding is not solely mediated by an electrostatic interaction between this anionic phospholipid and a region of positively net charge spanning W90, but also requires a binding-competent conformation of Hsp70, which is favored at low pH. Overall, the considerable impact of the mutation on the NBD at acidic pH explains the abolishment of key interactions of Hsp70 with BMP and the inability of the mutant to correct the dysfunctional lysosomal phenotype²⁴”

b. Line 191: “sufficient intensity”: please explain this a little bit more. It is unclear why this is the case.

Our response:

We thank the reviewer for pointing this out. The peptide did not have sufficient signal-to-noise ratio to allow an accurate estimation of the deuterium uptake. We hypothesize that BMP remained attached to the peptide, impairing the elution from the chromatographic column or impairing somehow the pepsin cleavage. To note, we are conducting the pepsin digestion and the chromatographic separation with solvents at acidic pH, a pH that favors binding. We now changed “sufficient intensity” into “sufficient signal-to-noise ratio” (line 190 with changes not shown), as this is a more accurate definition.

c. Line 198: please see comment 2a about PC being positive.

Our response:

We have changed “positive” into “zwitterionic”.

8. Page: 13

a. Line 265. (Please see also comment 7a). The sentence reads as if the mutation causes lipid-binding loss, but the results provided describe protein alterations that affect BMP binding.

Our response:

We thank the reviewer for highlighting again the lack of clarity and helping in improving the manuscript; we think we have addressed the reviewer’s concern with our answer to comment 7a.

b. Line 266. I understand that the site with the poly-His tag could not be mapped, but I am wondering how sure are we that this is binding? I am not arguing that at that pH the poly-His do not affect binding, I am just trying to understand it. Could these alterations be the result of the tag being so close to the binding site (and the region that changes by the pH shift) that causes these discrepancies?

Furthermore, the way the text is written, it implies that any tag (even a C-terminal) one would have the same effects. Is this true? If this is not, or it is not clear then maybe in addition to clarifying the above issues, it would be important to note everywhere in the text that this is an N-tagged protein.

Our response:

On line 256 (line 266 in previous version), we talk about “tag-induced binding effects”, as we observe certain effects in both N-terminal tagged proteins (WT and mutant). This could be due to a direct impact of the tag on the binding site (as they are very close) or due to the tag binding to the lipids, and, in turn, abrogating/decreasing the capability of other Hsp70 regions to bind. We have not stated in the manuscript that we found that the tag binds to the lipids, as that region cannot be mapped so we do not know. To accommodate the reviewer’s request of clarity, we have better clarified that we cannot determine whether the tag binds to lipids or not in the discussion (lines 357-358 with changes not shown). We think that a C-terminal his-tag would not have the same effect, therefore, to meet the reviewer’s request of clarification, we now added N-terminal His-tag in relevant parts of the text.

c. I am wondering what could explain the large differences in DeltaHDX values between the experiments presented in the different figures. What I am asking is: if we run this experimental setup and repeat it with different batches of protein will we get the same results? Will we see the same picks and the same levels of change or not? A clarification would be great here.

Our response:

HDX-MS experiments have, if conducted properly, quite low variability/error. This should also be apparent from our HDX-MS data on Hsp70 that includes technical replicates (n=3) that show an average SDs in the range of 0.04-0.06 D giving a rise to 98% CI values of typically around 0.4 D (e.g. see dotted gray lines in Fig 1-4). The difference in HDX between two states (in this case bound and unbound), which generates the peaks shown in the graph, would thus stay the same (within the above SD) for batches of Hsp70 for which the structure is the same. For our experiments, we have used the same protein batch as in Kirkegaard et al (2010), and this was done with the final aim to keep consistency between the papers, and thus our findings.

9. Page: 14

a. Line 301: I am not sure what this sentence means because the region that binds or does not bind was known. Is there something that I missed?

Our response:

We thank the reviewer for giving us the opportunity to clarify, as this is a key point of the manuscript. The binding of Hsp70 to BMP has been dissected at domain level (Mahalka et al (2014), Kirkegaard et al (2010), McCallister (2015)), but the literature currently lacks any information on more defined protein regions involved in the binding (we have introduced this in lines 58-60, changes not shown). Kirkegaard et al. (2010) have demonstrated a reduced binding affinity of Hsp70 upon mutation of W90, but the precise protein segments involved in the binding remain unknown. We have now mapped and discriminated, with the resolution of peptide segments, the regions interacting with zwitterionic lipids and/or BMP. Additionally, Kirkegaard et al. (2010) demonstrated that the mutant W90F lacks cytoprotective effect, but their data do not demonstrate that the interaction of this region to BMP directly induces the cytoprotective effect. By integrating our HDX data with the current literature, we think to have solved this ambiguity.

b. Line 306. Fig. S12: I am not sure how I can see the conformational properties in this figure. Please clarify if possible.

Our response:

We really thank the reviewer for spotting that. This is our mistake in referring to the figure. The figure we meant to refer to was S13, which shows the conformational dynamics of the protein. We have now corrected it in the text.

10. Page: 15

a. Line 317. It is not clear from the cited literature that HspA1A binds to the lysosomal membrane. Please clarify.

Our response:

We thank the reviewer for spotting this mistake in citing the literature. We have now referred to the following review (already cited among our references) where the binding of HspA1A to the lysosomal membrane is discussed in detail: Petersen, N. H. T., Kirkegaard, T., Olsen, O. D. & Jäättelä, M. Connecting Hsp70, sphingolipid metabolism and lysosomal stability. *Cell Cycle* 9, 2305-2309 (2010).

b. Line 325. Chaperone function alteration by pH: I am not sure I understand this finding. I am not arguing it is not true, just where does that come from?

Our response:

We take the reviewer's comment as an opportunity to clarify. We have described this finding in lines 120-124 (numbering with changes not shown), as we found that the main Hsp70 sites involved in the chaperone functions are highly destabilized (more than 15% of difference in HDX relative to the maximally labelled state) at lysosomal pH.

c. Line 331. HspA1A dimerization: Could this be supported by other lipid membranes in which Hsp70 was found to be a dimer or oligomer (sulfatide or phosphatidylserine)?

Our response:

We greatly appreciate the reviewer's suggestion to relate our findings to the current literature and we fully agree with the reviewer. We have now added the two citations that support our findings and take into account the reviewer's remark: Bacterial Hsp70 (DnaK) and mammalian Hsp70 interact differently with lipid membranes (ref. 54); Sulfatide-Hsp70 Interaction Promotes Hsp70 Clustering and Stabilizes Binding to Unfolded Protein (ref. 55)

11. Page: 16

a. Line 338. The idea of a lipid-binding pocket formation by pH change is great, but there is still binding at pH 7, albeit binding at lower pH is higher but still there is considerable binding at neutral pH. Could it be enhanced at these pH values because of the more pronounced conformational changes? Although H89 seems to be important, holistically the results do not seem to point to a residue but to a major conformational change of a region. What about other amino acids that are already protonated within this pH range (lysine or arginine)?

Our response:

We agree with the reviewer that the pH-induced increase in binding affinity is mainly due to a conformational maturation induced by pH. Indeed, as described in lines 321-324 (numbering with changes not shown), the region encompassing amino acids 85-122, which includes the most stabilized segment upon BMP interaction, is one of the regions showing the largest difference in HDX between pH 7.4 and 5.3. We have advanced the hypothesis that H89 plays a crucial role in inducing a conformation with higher binding affinity for BMP as this is the only residue within the segment 87-118 able to change its protonation status in the pH range considered (7.4-5.3), leading to a change of the protein intra-molecular electrostatic network and possibly also linked to observed structural re-arrangement. Histidine, whose side-chain pK_a is ~ 6 , has 0 net charge at cytosolic/extracellular pH (7.4) and becomes positively charged (approx. 90%) at endolysosomal acidic pH (5.3). As mentioned in the text (line 329), histidine residues often act as pH-sensors and mediate pH-dependent switches, therefore our hypothesis is supported by other examples in the literature. As the change in HDX between 7.4-5.3 is marked ($\Delta\text{HDX} > 15\%$) in region 85-118 and transmitted until aa 122, it appears that the pH-induced destabilizing effect is extensive, as the reviewer pointed out. The other charged amino acids mentioned by the reviewer are positively charged at both pH and it seems to us unlikely that they could act as triggers for a structural re-organization. In this regard, we have better clarified our hypothesis in lines 328-332 of the discussion.

b. Line 346 (please also refer to comments 7a and 13a). Is W90 responsible for binding or not?

Our response:

We have addressed this reviewer's concern with our answer to comment 7a.

c. Line 353 (please also refer to comments 2a and 7c) PC charge.

Our response:

We have changed "positive" into "zwitterionic".

12. Page: 17

a. Line 363 (please also refer to comment 8b). How sure are we that all poly-His tags will have the same effect? Maybe be specific about the N-terminal tag?

Our response:

We apologize in advance but we are not sure we understand the reviewer comment, as in line 363 (now line 349), we have clearly specified that this is an N-terminal hexahistidine tag and we have not stated anywhere (and we do not even think) that all poly-his tags would have the same effect. For instance, a C-terminal his-tag may not have the same effect. Anyway, we have added "N-terminal" in many places in the text when referring to the his-tag, following the reviewer's suggestion.

b. Line 373. Could this be because the flexible 6H tag is so close structurally to the "binding site?"

Our response:

Yes, we think so.

13. Page: 18

a. Line 392 (please also refer to comments 7a, 13a, and 16b). "The mutation impacts the dynamics of

the regions". If I am interpreting this sentence correctly then it might be changing the protein and not the binding, which also changes by the pH; is this correct? Please clarify.

Our response:

Based on our HDX results, we conclude that:

1. The pH change induces the binding competent conformation (fig. 1), and we postulate that this is triggered by the protonation of H89
2. The mutation impacts the dynamics of the N-terminal of the NBD, particularly at acidic pH (fig. 2, top plot), and it causes the loss of the binding competent conformation through the loss of two key hydrogen bonds (PDB 1S3X).
3. Indeed, the N-terminal region impacted by the mutation is also the region showing BMP specific effect in the WT (fig. 3 and 4, top plot), and the mutation changes the binding (fig. 4, bottom plot)
4. As a side note, the mutation changes the protein dynamics at neutral extracellular/cytosolic pH way less compared to the change at acidic pH (fig. 2, bottom plot) and in regions far from those implicated in the chaperone function, which supports the fact that the mutant retains the functional aspects of the Hsp70 chaperone (Kirkegaard et al, 2010).

b. Line 393. "Hydrogen bonds D69 and D86 with W90". I am not sure that I could find this information in the cited paper. Please clarify.

Our response:

We have cited the paper where the crystal structure of the NBD has been solved, the two hydrogen bonds are clearly indicated in the PDB file of the NBD structure (PDB 1S3X) solved in the paper. To clarify, we have now added the PDB code beside the citation.

14. Page: 19

Line 419. Figure 5. This is a great figure, but I was hoping to make it more informative by mapping the regions and adding the information of the N- or C- termini.

Our response:

We are very happy that the reviewer likes figure 5! With this figure, we aimed to illustrate the dual binding-mode of Hsp70 in different lysosomal environments, rather than mapping the regions involved in the interaction with the lysosomes. We have dedicated fig. 3B (non-tagged Hsp70) and in fig. 4B (N-his tagged Hsp70) to this detailed mapping. For these reasons, we have decided to make Fig. 5 more simple and schematic. Following the reviewer's suggestion, we have now indicated the regions involved in binding also here.

15. Page: 20

a. Line 443. Which clone did the authors use, which accession number?

Our response:

It is the same clone as in Kirkegaard et al, 2010.

b. Line 453. Which lipids used specifically? Source, company.

Our response:

We apologize for not having indicated the source of the lipids. The lipids have all been purchased from Avanti Polar-Lipids Inc. (Alabaster, AL, USA). We have now added this information in the method session.

c. Line 456. Would it be possible to provide the rationale of using the specific lipid mixtures? Why these specific lipids and why in these molar ratios?

Our response:

We thank the reviewer for giving us the opportunity to comment on that. There are two reasons why we have used these specific lipids and at these molar ratios:

1. Consistency with the liposome preparation made in Kirkegaard et al. (see caption of fig. 2d)
2. The lipid composition mimics that of lysosomal membranes; see following literature:
 - a. Escriba, P.V. et al. Membrane lipid therapy: Modulation of the cell membrane composition and structure as a molecular base for drug discovery and new disease treatment. *Prog Lipid Res.* 2015 Jul; 59:38-53.
 - b. G. van Meer, A.I. de Kroon. Lipid map of the mammalian cell. *J Cell Sci*, 124 (2011), pp. 5-8
 - c. G. van Meer, D.R. Voelker, G.W. Feigenson. Membrane lipids: where they are and how they behave. *Nat Rev Mol Cell Biol*, 9 (2008), pp. 112-124
 - d. S.E. Horvath, G. Daum. Lipids of mitochondria. *Prog Lipid Res*, 52 (2013), pp. 590-614

In order to meet the reviewer's request of clarification, we have added in the method session the reference to Kirkegaard et al. and the reference of Escriba et al. (ref 67, a review).

16. Page: 21

Line 464. Would it be possible to provide the rationale of using 47C?

Our response:

We have prepared our liposome with the method of the thin film hydration followed by extrusion. It is generally recommended to prepare the liposomes at 10°C above the lipid transition temperature; in the case of a lipid mixture, it is recommended to account for the lipid with higher transition temperature (sphingomyelin in our case).

- 16:0-18:1 PC (POPC): -2°C
- Sphingomyelin 37°C
- 18:1 BMP: Unknown; but as both hydrocarbon chains are unsaturated, we can predict a very low transition temperature (as for every other unsaturated phospholipids), and we are certain that this is lower than 37°C.

Reviewer 2:

The molecular chaperone Hsp70 is known to play a role in stabilizing the endo-lysosomal membranes. In this study, the authors follow up on their previous seminal studies in exploring the nature of interaction between the Hsp70 and lysosomal membranes. Hydrogen-deuterium exchange (HDX) MS is used to understand conformational changes of Hsp70 under different pH environments. Specific regions of Hsp70 that interact strongly with BMP are identified and the impacts of both the W90F

mutant as well as HIS tag are explored. Overall this study is rigorous and well-written, and contributes greatly to our understanding of the protective effects of Hsp70 in lysosomal storage disease. Specific comments:

1. Line 96-98

please provide a reference for the cut-off values for fast and slow HDX.

Our response:

We thank the reviewer for giving us the opportunity to clarify. There are no published community-agreed standard threshold values that define HDX kinetics as slow, medium or fast. After normalizing all the uptake values with that of the maximally labelled state (=100% uptake), we considered the >80% and <23% as suitable thresholds for distinguishing different types of HDX behavior. We now specifically state this definition in the section in question. These boundary values are generally in line with common practice in the field.

2. Line 118: Spelling mistake, should be "latter".

Our response:

We thank the reviewer for spotting this typo. We have now corrected.

3. Fig. 1: The graph axis shows a pH 5.3, but description mentions pH 5.4.

Our response:

We really thank the reviewer for spotting this and we have now corrected it.

4. Fig. 1: Please increase the font size on the x-axis "peptide number".

Our response:

We have now increased the size to improve readability.

5. For all graphs, please use "." Instead of "," for numerical values.

Our response:

We have now changed this throughout. Additionally, we have corrected a consistent mistake in reporting the measure unit for seconds (sec has been changed into s), and a consistent mistake of absence of space between numbers and measure units (the space has now been added everywhere).

6. Line 191-192: Mentions that residues 70-118, detectable in every analysis was not detected with sufficient intensity in the BMP state, which may represent a signature of strong lipid association. However, In the figure 3A (middle and lower panel) it does show some increase in HDX with BMP. Please could you comment on this?

Our response:

We thank the reviewer for the opportunity to clarify. We detected a decreased HDX in the BMP state in the peptide 85-118 (this is where the peak in the plot comes from), whereas peptides 69-84 and 69-86 did not show any difference in HDX. We agree that the text is a bit confusing here. We have now clarified with the following text (lines 187-192, with changes not shown):

“Two overlapping peptides (69-84 and 69-86) did not show change in HDX in the BMP-bound state, whereas peptide 85-118 manifested a significant decrease in HDX. The peptide comprising residues 70-118, detectable in every other analysis, was not detected with sufficient signal-to-noise ratio in the BMP state, which may represent a signature of strong lipid association even under the acidic quench conditions required by the HDX-MS workflow”

7. All the work presented is *in vitro*. It would be nice if the authors could comment on how well their findings may translate to a cellular context. For example, do the authors predict that co-chaperones of Hsp70 like DNAJA1 are involved in lipid binding given that they bind to Hsp70 near the proposed lipid binding region? In addition, Hsp70 is highly modified at the post-translational level (the chaperone code). We ask the authors to add the appropriate reference (Pubmed: 32518165) and discuss briefly the potential impact of PTMs on lipid binding. For example, there are several phosphorylated and acetylated amino acids in close proximity to W90 that may alter lipid binding. Finally, we would like the authors thoughts on why W90 is not conserved in other organisms (for example yeast Ssa1).

Our response:

We welcome the opportunity to elaborate on this topic. To the best of our knowledge, the binding of the bacterial homolog of the co-chaperone DNAJ on the bacterial homolog of Hsp70 (DnaK) occurs in the NBD subdomain IIA. The binding site of DNAJ to DnaK is on a region corresponding to residues ILTIDD (209-214) and FEVKATAG (217-224) of the human Hsp70. These residues are structurally distant from the main region of BMP binding (87-118). Although we cannot exclude partial overlap of the BMP and DNAJ binding sites in the lysosomal environment as Hsp70 structure is significantly altered under acidic conditions, our data do not seem to support the involvement of the co-chaperone in the lipid binding mechanism of Hsp70. We welcome the opportunity to comment on the impact of PTMs on lipid binding and the conservation of W90 in other organisms. We have now inserted two paragraphs in the discussion and included the requested references:

“Because of their critical role in cellular homeostasis, proteins of Hsp70 family are structurally and functionally conserved in evolution and found in both prokaryotes and eukaryotes⁶⁰. W90 is conserved in the human inducible Hsp70-2 (also called Hsp70-1b) and in the human constitutively expressed Hsc70⁶⁰. Nevertheless, W90 is, for instance, not conserved in the cytosolic Hsp70 isoforms of *S. cerevisiae* (Ssa1, Ssa2, Ssa3, and Ssa4), where both D69 and D86 are rather conserved. Notably, in Ssa1, Ssa2 and Ssa3, W90 is substituted with a Phe residue⁶¹. We know that W90F substitution does not abrogate Hsp70 chaperone function²⁴, however we observed that this mutation impairs the binding to BMP at lysosomal level. This suggests that human Hsp70 might have specifically evolved to stabilize the lysosomal membrane, a function probably absent in the other organisms, which, indeed, do not possess the same lysosomal machinery as humans. In this regard, both Hsp70-2 and Hsc70, despite the high sequence homology with Hsp70 and the conservation of W90, are not able to stabilize lysosomes²⁴. Hsp70-2 shares all but two amino acids (E110D, N499S) with Hsp70⁶⁰. Notably, E110 is comprised within the region identified as primary orchestrator of BMP interaction (87-118) and distant from H89 and W90, in support to a large protein segment participating in the association to BMP and/or in stabilizing the binding-competent conformation.” (lines 382-397)

“We note that the Hsp70 protein used in this work is expressed in bacteria and thus without native human post-translational modifications (PTMs). However, *in vivo*, proteins of Hsp70 family can be highly modified at post-translational level to regulate localization, refolding activity and interaction with cognate partners⁶¹. In the human Hsp70, there are several sites found with PTMs in proximity to

W90, including phosphorylation of H89⁶¹, which may impact (or regulate) the BMP interaction of this region.” (lines 410-416).

Reviewer 3:

In the manuscript by Calvaresi et al. more structural information is provided about the interaction of the heat shock protein 70 (hsp70) to the anionic bis-(monoacylglycerol)-phosphate (BMP) usually present in the membranes of intraluminal vesicles in lysosomes. The authors build up on earlier studies where this binding was defined to occur within the nucleotide binding domain (NBD) of Hsp70 and a Hsp70 W90F mutant was reported to have almost lost interaction to BMP (e.g. ref#24). The current study makes use of hydrogen-deuterium exchange mass spectrometry analysis. The knowledge about the importance of the NBD and the relevance of the Hsp70W90F mutant for mediating the BMP binding is confirmed. A lipid binding dual mode model is suggested which apparently is pH dependent, i.e. dependent on the subcellular localization (late endosomes vs lysosomes). At pH5.3 Hsp70 exerts a native fold and NBD binds to the membrane and the substrate binding domain is more peripherally localized. At a more lysosomal pH of 4.5 the authors suggest a globule conformation and interactions to the BMP-containing membrane mediated by unspecific electrostatic interactions.

Although the manuscript is clearly written I doubt that the new structural insights reported here go far beyond the knowledge which was already available. It remains a bit vague in how far the hydrogen-deuterium exchange mass spectrometry data can be translated in a more physiological set up, e.g. by addressing selective Hsp70 mutations and analyzing lysosomal functions under different conditions. At the present stage this study seems more appropriate for a specialized journal. I would have also expected -at least for this journal- experiments (structure-function analyses, CRISPR knockin cell lines) which address the suggested model in suitable cell based assays (e.g. in NPC1 KO cells).

Our response:

We thank the reviewer for appreciating our work, even though it did not fully satisfy his/her expectations. Unfortunately, the methodologies suggested by the reviewer to expand our work with cell-based assays are not available to us. Also, we note that the use of mutagenesis to study Hsp70 function also is not without its limitations as such mutations may cause undetected yet undesired non-native changes to the delicate dynamics of Hsp70. Please note that in the present work we actually detect such significant changes to Hsp70 conformation upon mutagenesis (W90F) and even the addition of an N-terminal His-tag. Such changes were not known to occur before and require a technique like HDX-MS to detect. We thus believe that one of the premier merits of the current work is that we use the HDX-MS technique which has allowed us to provide some of the first detailed insights into the conformational properties and lipid binding of the wildtype Hsp70 under very native-like solution conditions, in addition to the W90F mutant. Furthermore, we note that the lysosomal function of Hsp70 in cell-based assays has already been investigated with Hsp70 trans-genic murine embryonic fibroblasts (Hsp70-MEFs) and in fibroblasts from a patient with NPD A (in the latter experiments, both WT and W90F Hsp70 were tested) (Kirkegaard et al, 2010).

In summary, below we have highlighted the merits and new aspects of our work (from a biological and methodological point of view) and why we are fully convinced that the manuscript would be of interest to the readers of Communications Biology:

1. Our work provides the first detailed comparison of the native solution-phase conformational properties of human Hsp70 at physiological pH (7.4) and the pHs of late endosomes (5.3) and mature lysosomes (4.5). Furthermore, we provide the first detailed mapping (at a resolution of short amino acid segments) of the regions of Hsp70 that interact with the inner lysosomal membrane. Importantly, we are able to distinguish and map the specific binding impact of BMP on Hsp70 from the effects of non-specific binding of Hsp70 to other major lipid components of the lysosomal membrane. In the current literature the lipid binding of Hsp70 has so far only been resolved at the domain level, describing the involvement of the NBD and/or SBD, without confining the binding sites to any determined region. Finally, our study allows us to propose, for the first time, the region 87-118 to be the region of Hsp70 directly responsible for mediating the cytoprotective effect.
1. The interaction of Hsp70 to the lysosomal membrane is part of the mechanism through which Hsp70 reverts Niemann-Pick diseases, currently lacking effective treatments. Hsp70 is a drug candidate and a therapeutic strategy based on amplification of Hsp70 is currently being evaluated in late-stage clinical trials. Therefore, we think that the data presented here can aid in the understanding of the mechanism of action of Hsp70 in reverting LSDs and in drug design, potentially being of great interest in the near future for the scientific community.
2. Communications Biology has recently showed an interest toward the HDX-MS technique (Salmas RE, Borysik AJ. *Commun Biol.* 2021 Feb 15;4(1):199, HDXmodeller: an online webserver for high-resolution HDX-MS with auto-validation). Despite the fact that the use of HDX-MS has expanded significantly in recent years, few HDX-MS-based works published so far focus on the study of protein-lipid interactions. This is due to the technical challenges that working with membrane proteins and lipids entail, and also due to the fact that protein-lipid binding sites, especially for peripheral membrane proteins, are notoriously very difficult to study using traditional methods in structural biology. We have successfully overcome these challenges with an extensive HDX-MS method optimization, clearly detailed in the Supplementary Information (how we mitigated the detrimental effect of BMP on the LC-MS analysis) and method session (how we designed the deuterium labelling to study an interaction with k_d in the micromolar order, transient and mainly electrostatic, hence very difficult to capture by differential HDX). Therefore, we reckon that this manuscript can be highly interesting also from a technical/method-based viewpoint.

Additional comments:

2. More (experimental and literature) information is required how and under which conditions a cytosolic protein (hsp70) may enter the lumen of endosomes and lysosomes. How is the translocation of hsp70 from the cytosol to endocytic vesicles mediated and what is the fate of hsp70 in such an enigmatic translocation process?

Our response:

We thank the reviewer for the opportunity to elaborate. *In vivo*, Hsp70 and other Heat Shock Proteins are known to be present in the extracellular space, in particular in response to various stresses (De Maio 2014, DOI: 10.2174/1389203715666140331113057). As described in reference 29 (Kirkegaard et

al., 2016) and reference 30 (Petersen et al., 2010, Cell cycle), extracellular Hsp70 is thought to enter the lysosomal compartment through receptor-mediated endocytosis and subsequent fusion of late endosomes with the lysosomal compartment.

We have now added this information to the introduction (lines 35-38, with changes not shown):

“In vivo, Hsp70 and other heat shock proteins are known to be present in the extracellular space, in particular in response to various stresses^{27,28}. Extracellular Hsp70 can enter the endo-lysosomal compartment through receptor-mediated endocytosis and subsequent fusion of late endosomes with lysosomes^{29,30}.”

3. Is the BMP found at the inner side of the limiting membrane of lysosomes or as reported more abundantly within intraluminal vesicles of late endosomes? To which membrane subdomain does hsp70 bind to and what is the dynamics of such a binding process? How can hsp70 overcome the glycocalyx barrier which is present at the inner membrane of lysosomes. Experiments performed in liposomes do not necessarily represent a very physiological situation.

Our response:

We thank the reviewer for the comment and can refer to the available literature. When present in late endosomes that fuse into lysosomes, Hsp70 is exposed to the intraluminal lipid vesicle in the lysosome, and this is the main site of binding to BMP (ref. 30: Petersen et al., 2010, Cell cycle); BMP is not present in the limiting lysosomal membrane, but is present more abundantly in the intraluminal vesicles membrane (ref. 26: Kolter and Sandhoff 2009, FEBS letters; ref 30: Petersen et al., 2010, Cell cycle).

For clarity, we have now changed “inner membrane” into “membrane of the intraluminal vesicles” (lines 34-35) and detailed more clearly where Hsp70 binds (line 38-39, with changes not shown). As described in our answer to comment 2, Hsp70 enters the endosomes by endocytosis and subsequently enters lysosomes when endosomes fuses with lysosomes. Concerning our membrane mimetic model, i.e. liposomes, we would like to underline that the size of the intraluminal lysosomal vesicles are 30–120 nm (ref. Tancini et al., 2019; doi: 10.3390/genes10070510), thus comparable to the diameter of our liposomes, which are likewise also highly monodisperse (see table 7, Supplementary Information). This means that the curvature of our liposomes mimics that of the lysosomal intraluminal vesicles. The composition of our liposomes, in terms of lipid components, also accurately mimics that of native lysosomal membranes (see also our answer to Comment 15c-line 456 of Reviewer 1). Additionally, all the studies published so far on the interaction between Hsp70 and lipids/BMP have been carried out with liposomes, both to explore Hsp70-membrane interaction in an acidic and neutral environment. See some examples: (Mahalka et al (2014); Kirkegaard et al (2010); McCallister (2015) doi.org/10.1016/j.bbrc.2015.10.057; McCallister (2015) 10.1038/srep09363; Arispe et al. (2002); Armijo et al. (2014); Lopez et al. (2016). We thus think that our membrane mimetic model are highly suitable for the purpose of our study.

4. Can the critical role of the region between hsp70 aa87-118 more precisely defined and can selected mutants be tested in a suitable set up to restore lysosomal storage (in NPC-deficient cells)? More specifically: the critical H89 residue which is adjacent to the W90 residue seems to trigger the structural rearrangement into a BMB binding-competent conformation. How can this be explained? Further validation by cell-based assays are required. A possible dimerization of Hsp70 is discussed but not experimentally addressed. Dimerization may well affect binding to BMP and requires more attention

Our response:

We propose that H89 plays a crucial role in inducing a conformation with higher binding affinity for BMP as this is the only residue within the region 87-118 able to change its protonation status in the pH range considered (7.4-5.3), leading to a change of the protein intra-molecular electrostatic network and a structural re-arrangement. As mentioned in the text (line 329, with changes not shown), histidine residues often act as pH-sensor regulators and mediate pH-dependent switches, and our hypothesis seems to be supported by other examples in the literature.

While agreeing that additional experiments including mutagenesis of H89 and a development of a cell-based assay could bring additional insight into the specific binding mechanism between HSP70 and BMP, we have chosen to focus on the mutation that was previously shown to directly affect the physiological effect of Hsp70 in the context of lysosomal disease (Kirkegaard et al., 2010). Also, we note that a strategy of mutagenesis of H89 is likely to incur non-native changes to the delicate dynamics of Hsp70 and thus not be useful and would, in any case, require extensive biophysical characterization work in addition to the actual cell-based assay. In light of the extensive body of highly informative labor-intensive experimental data already provided using HDX-MS, we believe such additional mutagenesis data to be beyond the scope of the present work. We hope the reviewer can appreciate this viewpoint.

Concerning Hsp70 dimerization at acidic pH, we observed that a possible dimerization is triggered by the acidic pH (in the region 520-544, containing sites of dimerization already described in the literature), and the dimer would be the membrane bound form of the protein. Indeed, our data show that there is no difference in HDX in region 520-544 between the unbound and membrane-bound state of the protein (fig. 3A and fig 4A). Therefore, assuming our hypothesis being correct, this dimerization phenomenon does not appear to be affecting BMP binding. As pointed out by reviewer 1 (comment 10, page 15.c. Line 331), our hypothesis is supported by other evidences in the literature, where Hsp70 was found to dimerize/oligomerize at the interface with the membrane.

5. The authors mention (line 76) a BMB interaction in the cytosol. Where should this occur?

Our response:

We thank the reviewer for spotting this mistake. Indeed, BMP is abundant in the lysosome membrane but generally not present in the membrane of other cell compartments. We have now corrected the sentence with "...and explain its low affinity for BMP at neutral pH" (line 75, with changes not shown). This relates to the current literature where Hsp70 is described to weakly bind at neutral pH and to have increased binding affinity for this phospholipid under acidic conditions.

REVIEWERS' COMMENTS:

Reviewer #1 (Remarks to the Author):

The revised manuscript addresses appropriately most comments raised by the reviewers. The results are new, important, and compelling.

I have two comments (stemming from the revised text triggered by the other reviewer comments).

1. "the high sequence homology with Hsp70 and the conservation of W90," (line 463). The two proteins are homologous, please change the word homology (qualitative) to identity (quantitative).
2. "In vivo, Hsp70 and other heat shock proteins are known to be present in the extracellular space, in particular in response to various stresses^{27,28}. Extracellular Hsp70 can enter the endo-lysosomal compartment through receptor-mediated endocytosis and subsequent fusion of late endosomes with lysosomes^{29,30}".

I am not sure we know this for a fact. What we know and I apologize if I am wrong, is that recombinant Hsp70 enters the cells and ends up in lysosomes. I am not sure we know for receptor-mediated endocytosis and all the other details written. I think this needs to be re-written to be consistent with what we know. Furthermore, the finding that Hsp70 when administered in cells does that may not necessarily be true for native Hsp70 (which has been suggested to use an endo-lysosomal route to secrete).

Reviewer #2 (Remarks to the Author):

The authors have put substantial effort into addressing all reviewer concerns and congratulate the authors on producing a substantially improved manuscript. I highly recommend that this interesting and novel study be published in its current form.

Andy Truman

Reviewer #1 (Remarks to the Author):

The revised manuscript addresses appropriately most comments raised by the reviewers. The results are new, important, and compelling.

I have two comments (stemming from the revised text triggered by the other reviewer comments).

1. "the high sequence homology with Hsp70 and the conservation of W90," (line 463). The two proteins are homologous, please change the word homology (qualitative) to identity (quantitative).

Our response: We thank the reviewer for pointing out this and have now changed "homology" into "identity".

2. "In vivo, Hsp70 and other heat shock proteins are known to be present in the extracellular space, in particular in response to various stresses^{27,28}. Extracellular Hsp70 can enter the endo-lysosomal compartment through receptor-mediated endocytosis and subsequent fusion of late endosomes with lysosomes^{29,30}".

I am not sure we know this for a fact. What we know and I apologize if I am wrong, is that recombinant Hsp70 enters the cells and ends up in lysosomes. I am not sure we know for receptor-mediated endocytosis and all the other details written. I think this needs to be re-written to be consistent with what we know. Furthermore, the finding that Hsp70 when administered in cells does that may not necessarily be true for native Hsp70 (which has been suggested to use an endo-lysosomal route to secrete).

Our response: We thank the reviewer for the pertinent comment and acknowledge that there may be differences in the biological responses between recombinant Hsp70 and extracellular endogenous Hsp70 arising as a physiological response to stress, and have rewritten the sentence accordingly.

"In vivo, in particular in response to various stresses, Hsp70 and other heat shock proteins are known to be released in the extracellular space, where they elicit various biological responses facilitated by their binding to a number of receptors from different families, such as the toll-like receptors (TLR), scavenger receptors and c-type lectins²⁷⁻²⁹. A cellular uptake mechanism for extracellular Hsp70, by which it may enter the endo-lysosomal compartment, has been demonstrated for recombinant Hsp70 through its interaction with the scavenger receptor LRP-1/CD91^{30,31}."

We added a new reference, ref. 29: Calderwood, S. K., Theriault, J., Gray, P. J., Gong, J. Cell surface receptors for molecular chaperones. *Methods* 43, 199-206 (2007).